# Specification of oxytocinergic and vasopressinergic circuits in the developing mouse brain

María Pilar Madrigal [1✉] & Sandra Jurado [1✉]

Oxytocin (OXT) and arginine vasopressin (AVP) support a broad range of behaviors and homeostatic functions including sex-specific and context-appropriate social behaviors. Although the alterations of these systems have been linked with social-related disorders such as autism spectrum disorder, their formation and developmental dynamics remain largely unknown. Using novel brain clearing techniques and 3D imaging, we have reconstructed the specification of oxytocinergic and vasopressinergic circuits in the developing mouse brain with unprecedented cellular resolution. A systematic quantification indicates that OXT and AVP neurons in the hypothalamus display distinctive developmental dynamics and high cellular plasticity from embryonic to early postnatal stages. Our findings reveal new insights into the specification and consolidation of neuropeptidergic systems in the developing CNS.

[1] Instituto de Neurociencias CSIC-UMH, San Juan de Alicante, Alicante, Spain. ✉email: mmadrigal@umh.es; sjurado@umh.es

Oxytocin (OXT) and arginine vasopressin (AVP) are evolutionarily conserved neuropeptides implicated in the regulation of complex social behaviors. These neuropeptides are synthesized in the hypothalamus where are processed into a nine amino acid peptide chain only differing in two amino acids[1]. OXT was first identified for its systemic function in parturition and lactation[2–4]. Additionally, cumulative evidence indicated a prominent role of OXT in parental and copulatory behaviors[5–7] and as a pro-social neuromodulator[8,9] increasing the salience of social stimuli and regulating social behaviors as well as aggression, anxiety, fear, and trust[10–20]. Conversely, AVP has been proposed to antagonize OXT-mediated functions[21,22] like increasing social stress[23] and reducing interpersonal trust in humans[23–25]. However, this view is being revisited after several studies exposed a more complex scenario for the interplay of OXT and AVP in the central nervous system (CNS)[5,6,8,9,26]. Although the details of how OXT and AVP interact to modulate neuronal function remain unknown, numerous studies have revealed that alterations of OXT and AVP circuits may underlie mental disorders often characterized by deficits in social interaction such as autism[27–31], social anxiety and aggression[26,32], and schizophrenia[33–35].

Previous work suggests that the balance between OXT- and AVP-mediated signaling is likely to determine the display of appropriate social behaviors, thus understanding their simultaneous developmental dynamics is crucial for having a complete picture of their regulatory roles. Most studies characterizing the expression of OXT and AVP projections have employed histological methods and in situ hybridization in fixed sections[36–39] which provide insightful information, but are difficult to extrapolate to circuit formation in the entire brain. Furthermore, most of the previous work has focused on the rat brain[40–46] even though an increasing number of studies employ the mouse as an experimental model, thus highlighting the need for more accurate connectivity maps for this commonly used species.

To overcome these current challenges, we have now implemented the iDISCO+ clearing technique to analyze the expression of endogenous neuropeptides. This method in combination with light sheet fluorescent microscopy has allowed us to generate 3D reconstructions of the oxytocinergic and vasopressinergic systems in the entire mouse brain from early development to adulthood. This methodology achieves cell-specific resolution allowing a systematic quantification of oxytocinergic and vasopressinergic cells. Our data revealed that OXT and AVP neurons first appear at the caudal hypothalamic regions followed by rostral areas. Interestingly, different hypothalamic nuclei show marked differences between OXT and AVP expression during development. The rostral nuclei, such as the anterodorsal preoptic nucleus (ADPN) and the periventricular nucleus (PeVN), together with neighboring areas such as the bed nucleus of stria terminalis (BNST) display a fairly homogeneous expression whereas the caudal nuclei (supraoptic nucleus (SON), paraventricular nucleus (PVN), and the retrochiasmatic area (RCH)) exhibit greater heterogeneity with a large population of neurons coexpressing both OXT and AVP during early postnatal stages. Our analysis indicates that the accessory nucleus (AN) is the most heterogeneous of the hypothalamic nuclei with a rostral section almost exclusively constituted by OXT and OXT + AVP neurons and a caudal region enriched with AVP neurons.

Although the expression of OXT + AVP neurons is quite high at postnatal day seven (PN7), most of the nuclei exhibit an increase in the number of OXT neurons in detriment of the mixed OXT + AVP population during postnatal development. This switch in neuropeptide expression is particularly significant in BNST and ADPN, where the presence of AVP neurons decreases drastically in the adult brain. These developmental adaptations are

expected to have functional consequences impacting the ratio of AVP/OXT innervation to their projection sites.

In addition to wild type animals, we have characterized OXT expression in a commercial OXT-Cre mouse line[47]. These studies have confirmed our observation of a significant increase of oxytocinergic neurons in most hypothalamic nuclei as development progresses. Strikingly, some OXT positive neurons in the SON and RCH of the commercial OXT-Cre line are not recognized by standard OXT antibodies suggesting the presence of nucleus-specific OXT precursors.

In summary, our in-depth circuit analysis has revealed that OXT and AVP expression exhibits distinct developmental dynamics in the mouse brain. These dynamic adaptations are likely to modulate the functional properties of different brain regions according to their developmental stage, thus contributing to the refinement of the neuronal circuits that support context-appropriate social behaviors later in life.

## Results

**Application of iDISCO+ for identifying OXT and AVP hypothalamic circuits.** Understanding the formation and developmental dynamics of oxytocinergic and vasopressinergic circuits is of great significance given their regulatory role in social behaviors. The identification of alterations during the development of these systems is expected to reveal critical insights into the underlying causes of several neuropsychiatric disorders[19–35]. Most of OXT and AVP innervation in the CNS comes from the hypothalamus, a brain area comprising highly heterogeneous nuclei. Pioneer work using in situ hybridization and immunohistochemistry techniques described with great detail the hypothalamic circuit in the developing rat brain[40–46], however, the mouse brain attracted less attention partly because of its smaller size which hinders the identification of the intricate hypothalamic nuclei. Because an increasing number of studies particularly on social behavior utilize mouse models, it is necessary to improve our understanding of the development of OXT and AVP circuits in this species.

Recent advances on tissue clearing techniques have enabled the study of neuronal connectivity in the whole brain by means of light sheet microscopy and 3D reconstructions[48–50]. Here, we implemented a protocol for 3D imaging of solvent-cleared brains, adapted from the iDISCO+ protocol described in Renier et al.[51]. Incubation time for anti-OXT and anti-AVP antibodies was optimized according to brain size and tissue firmness. Best results were obtained with incubation times of 5 days for E16.5 brains, 10 days for PN0, 2 weeks for PN7, and 3 weeks for adult brains. In our hands, these modifications resulted in fully cleared brains (Fig. 1a) with very low background which allowed to unambiguously distinguish single cells in different hypothalamic nuclei and neighboring brain areas (Fig. 1b, Supplementary movie 1) including (from rostral to caudal): BNST (bed nucleus of stria terminalis), PeVN (periventricular nucleus), VMPO (ventromedial preoptic nucleus), ADPN (anterodorsal preoptic nucleus), SON (supraoptic nucleus), SCH (suprachiasmatic nucleus), PVN (paraventricular nucleus), RCH (retrochiasmatic area) and AN (accessory nucleus, constituted by scattered cells within the hypothalamic area not included in any of the other nuclei[52]). This method in combination with 3D image processing, enabled isolation of single hypothalamic nuclei (surface Imaris tool; Fig. 1c), and the analysis of different neuronal populations within each nucleus (spots Imaris tool; Fig. 1d). The technique additionally allowed us to identify a small population of OXT and AVP positive cells in other brain areas such as the medial amygdala (MeA), where OXT and AVP positive neurons were mainly found in the anteriodorsal and anteroventral part (Table 1). These results indicate that iDISCO+ provides an

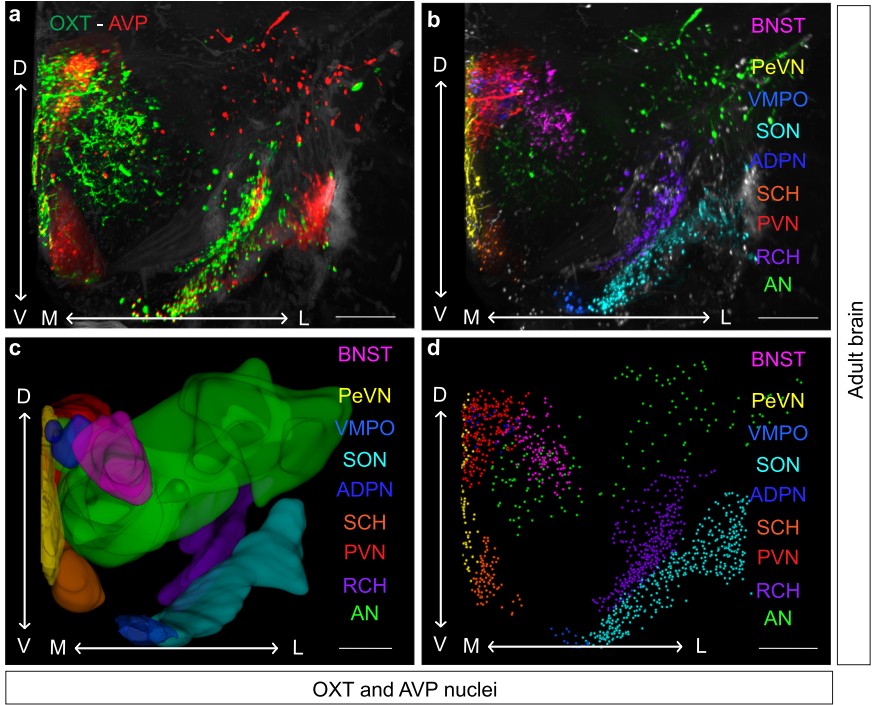

**Fig. 1 iDISCO+, a useful tool for distinguishing between nuclei.** Representative image of the hypothalamus of one hemisphere of an adult mouse brain processed with the iDISCO+ protocol and stained for OXT (green) and AVP (red) (**a**) in which distinct neuronal populations can be easily resolved. The image shows the hypothalamic area displaying oxytocinergic and vasopressinergic neurons pseudocolored in each hypothalamic nucleus and surrounding areas such as the BNST (**b**). OXT and AVP expression in each nucleus are represented using the Imaris surface tool (**c**). OXT and AVP neurons in each nucleus are identified and outlined for performing quantitative analysis (**d**). Arrows: D dorsal; V ventral; M medial; L lateral. PeVN periventricular nucleus, ADPN anterodorsal preoptic nucleus, BNST bed nucleus of stria terminalis, VMPO ventromedial preoptic nucleus, SON supraoptic nucleus, PVN paraventricular nucleus, RCH retrochiasmatic area, AN accessory nucleus, SCH suprachiasmatic nucleus. Scale bar: 300 μm.

excellent method for preserving the identity of peptides like OXT and AVP, and permits the identification of intricate nuclei such as hypothalamic structures.

**Different hypothalamic nuclei exhibit distinct temporal windows of OXT and AVP expression.** The precise anatomical resolution achieved by iDISCO+ allowed the isolation of individual hypothalamic nuclei for quantification of both OXT and AVP neurons at a variety of developmental stages (see Supplementary movies 2–5 for 3D whole-brain reconstructions). Our analysis revealed significant differences in the expression dynamics of OXT and AVP in the developing mouse hypothalamus. Whereas OXT and AVP positive neurons could be identified at caudal nuclei like the PVN, SON, and RCH as early as E16.5, rostral hypothalamic nuclei remained unlabeled (Fig. 2a–d, Supplementary Fig. 1, and Tables 1 and 2). However, at PN0 all the hypothalamic nuclei showed neurons expressing OXT and AVP with the exception of the SCH which expresses AVP cells almost exclusively[53] (Fig. 2e–h and Supplementary Fig. 1). OXT and AVP expression patterns were similar in all hypothalamic nuclei from early postnatal to adulthood (Fig. 2i–p). These data indicate that OXT and AVP neurons first appear in caudal areas, in agreement with the role of these nuclei as main sources of OXT and AVP modulation (Fig. 2 and Supplementary Fig. 1)[54]. As a general rule, the number of OXT and AVP neurons increases over development in all hypothalamic areas (Table 1). This proliferation is likely to occur in parallel with brain growth and its increasing demands for neuromodulatory innervation.

Temporal and spatial heterogeneity of OXT and AVP expression in different hypothalamic nuclei was revealed by flatmap representations of 3D brain reconstructions (Fig. 3).

Hypothalamic flatmaps convey the intensity of OXT and AVP positive cells providing a quantitative representation of cell density in each nucleus. 3D-based hypothalamic flatmaps showed clear differences between nuclei in the expression pattern of OXT and AVP during development (see density quantifications in Supplementary Fig. 2). Remarkably, whereas the density of OXT positive neurons seems to steadily increase (Fig. 3e–h), AVP neurons show the opposite behavior (Fig. 3i–l). Interestingly, neurons co-expressing OXT and AVP show an increase at early postnatal stage (PN0) (Fig. 3m–p), coinciding with a critical period for the maturation of social behaviors.

**Neurons co-expressing AVP and OXT are a common feature of the developing hypothalamus.** Cell quantification confirmed that AVP and OXT systems in distinct hypothalamic nuclei differ in their temporal dynamics (Fig. 4). The percentage of AVP neurons is higher earlier in life and progressively declines during adulthood in most of the nuclei (Figs. 4 and 5, Table 2, percentage of AVP neurons normalized to total number of cells: E16.5: 42.62 ± 2.78; PN0: 52.25 ± 6.04; PN7: 27.97 ± 3.11; Adult: 27.43 ± 3.65; n = 4 per each developmental stage, mean ± S.E.M). This reduction is also observed in AVP-dominant nuclei like the SCH[53] where the percentage of AVP-expressing cells decreases from early postnatal stages (PN0) to adulthood (Supplementary Fig. 3 and Fig. 5). Most nuclei show a similar reduction in the percentage of AVP cells with maturation, with the exception of the PVN that exhibits a non-significant increase of vasopressinergic neurons in the adult brain (Fig. 4w and Table 2).

Quantification of 3D imaging revealed that at early postnatal stages most hypothalamic nuclei exhibit a high percentage of neurons co-expressing OXT and AVP (Figs. 4f–o and 5, Tables 1 and 2;

**Table 1 Total number of OXT, AVP, and OXT + AVP neurons in the hypothalamus nuclei during development.**

| OXT | E16.5 | PN0 | PN7 | Adult |
|---|---|---|---|---|
| BNST | 0.00 | 2.00 ± 2.00 | 16.00 ± 9.94 | 100.00 ± 36.46 |
| PVN | 24.50 ± 14.22 | 123.75 ± 33.62 | 131.25 ± 70.70 | 461.25 ± 20.60 |
| SON | 116.75 ± 12.60 | 33.00 ± 13.37 | 83.50 ± 50.49 | 216.00 ± 44.67 |
| VMPO | 0.00 | 6.25 ± 2.46 | 3.25 ± 3.25 | 8.75 ± 4.84 |
| RCH | 99.25 ± 11.92 | 63.25 ± 16.85 | 70.75 ± 8.62 | 101.75 ± 28.05 |
| ADPN | 0.00 | 0.00 | 17.75 ± 14.79 | 75.00 ± 28.67 |
| PeVN | 0.00 | 30.25 ± 18.22 | 50.50 ± 31.62 | 123.50 ± 45.17 |
| AN | 0.00 | 37.5 ± 15.76 | 11.50 ± 5.45 | 164.75 ± 18.28 |
| SCH | 0.00 | 3.67 ± 1.61 | 0.67 ± 0.58 | 4.00 ± 1.78 |
| MeA | 0.00 | 0.25 ± 0.25 | 0.25 ± 0.25 | 0.50 ± 0.50 |

| AVP | E16.5 | PN0 | PN7 | Adult |
|---|---|---|---|---|
| BNST | 0.00 | 2.75 ± 1.80 | 9.50 ± 3.80 | 1.5 ± 0.65 |
| PVN | 61.5 ± 8.87 | 138.25 ± 33.69 | 88.00 ± 59.30 | 206.50 ± 74.77 |
| SON | 129 ± 6.36 | 122.25 ± 12.64 | 207.25 ± 17.67 | 169.50 ± 45.75 |
| VMPO | 0.00 | 4.00 ± 0.58 | 2.00 ± 1.22 | 2.25 ± 0.48 |
| RCH | 107.50 ± 20.23 | 73.00 ± 14.70 | 128.00 ± 40.19 | 69.75 ± 8.54 |
| ADPN | 0.00 | 3.25 ± 2.93 | 3.25 ± 2.29 | 2.00 ± 0.71 |
| PeVN | 0.00 | 11.50 ± 4.33 | 56.5 ± 31.47 | 14.5 ± 3.28 |
| AN | 0.00 | 85.00 ± 10.87 | 92.00 ± 16.91 | 95.00 ± 12.51 |
| SCH | 0.00 | 368.67 ± 24.50 | 322.67 ± 27.57 | 175.25 ± 27.68 |
| MeA | 0.00 | 0.00 | 2.75 ± 1.38 | 0.00 |

| OXT-AVP | E16.5 | PN0 | PN7 | Adult |
|---|---|---|---|---|
| BNST | 0.00 | 53.75 ± 8.09 | 157.50 ± 11.91 | 53.50 ± 49.58 |
| PVN | 130.75 ± 16.27 | 68.75 ± 31.07 | 388.00 ± 72.50 | 93.50 ± 24.13 |
| SON | 13.50 ± 2.72 | 51.50 ± 18.31 | 168.75 ± 40.24 | 78.75 ± 29.56 |
| VMPO | 0.00 | 8.00 ± 2.80 | 14.50 ± 3.07 | 21.00 ± 12.92 |
| RCH | 11.75 ± 1.25 | 63.75 ± 23.67 | 86.00 ± 32.54 | 50.00 ± 4.42 |
| ADPN | 0.00 | 23.50 ± 6.96 | 58.75 ± 23.57 | 17.00 ± 14.02 |
| PeVN | 0.00 | 10.25 ± 4.25 | 91.00 ± 14.87 | 57.00 ± 37.41 |
| AN | 0.00 | 24.50 ± 14.15 | 148.25 ± 11.52 | 15.50 ± 4.17 |
| SCH | 0.00 | 0.67 ± 0.58 | 2.33 ± 0.29 | 3.00 ± 1.47 |
| MeA | 0.00 | 0.50 ± 0.50 | 7.50 ± 1.19 | 3.00 ± 1.78 |

Quantification of total number of each neuron population for different developmental stages (E16.5, PN0, PN7, and adult brain).
Results are expressed as mean ± S.E.M. (n = 4 brains).
*BNST* bed nucleus of stria terminalis, *PVN* paraventricular nucleus, *SON* supraoptic nucleus, *VMPO* ventromedial preoptic nucleus, *RCH* retrochiasmatic area, *ADPN* anterodorsal preoptic nucleus, *PeVN* periventricular nucleus, *AN* accessory nucleus, *SCH* suprachiasmatic nucleus, *MeA* medial amygdalar nucleus.

percentage of cells from the whole hypothalamus combining all nuclei E16.5: OXT = 34.66% ± 3.21; AVP = 42.62% ± 2.78; OXT + AVP = 22.72% ± 2.75; PN0: OXT = 20.93% ± 4.91; AVP = 52.25% ± 6.04; OXT + AVP = 26.83% ± 10.74; PN7: OXT = 12.91% ± 4.41; AVP = 27.97% ± 3.11; OXT + AVP = 39.39% ± 6.97; Adult: OXT = 52.00% ± 5.10; AVP = 27.43% ± 3.65; OXT + AVP = 17.25% ± 5.91; $n = 4$ per each stage). This phenomenon is quite prominent in some nuclei, such as ADPN and neighboring areas like the BNST, where the percentage of neurons co-expressing OXT and AVP decreases in the adult brain, although in a rather heterogeneous fashion (Figs. 4x–y, 5 and Table 2, percentage of AVP + OXT neurons: PN0: ADPN = 88.54% ± 7.86; BNST = 90.16% ± 8.34; PN7: ADPN = 77.91% ± 8.45; BNST = 87.68% ± 5.95; Adult: ADPN = 26.50% ± 22.93; BNST = 26.55% ± 24.07; $n = 4$ per each stage). In the rest of the nuclei, co-labeling of OXT and AVP reaches a peak at PN7 that thereafter steadily decreases over time (Fig. 4u–w and Supplementary Fig. 3q–s), with the exception of SCH which is primarily constituted by AVP-expressing neurons (Supplementary Fig. 3t and Fig. 5, Tables 1 and 2; SCH neurons: PN0: OXT: 0.69% ± 0.40; AVP = 98.93% ± 0.48; OXT + AVP = 0.12% ± 0.12; PN7: OXT = 0.16% ± 0.16; AVP = 99.08% ± 0.14; OXT + AVP = 0.54% ± 0.18; Adult: OXT = 2.22% ± 0.86; AVP = 96.05% ± 1.22; OXT + AVP = 1.73% ± 0.79; $n = 4$ per each stage).

In contrast, OXT neurons exhibit the opposite trend with their lowest expression at PN7 from when OXT significantly increases to reach its maximum levels in the adult brain (Fig. 4). The transition from OXT + AVP to OXT expression is more pronounced in the BNST and ADPN where the percentage of AVP neurons is drastically reduced in the adult brain, although with some variability as previously observed[39] (Fig. 4x and y). These results indicate that neurons co-expressing OXT and AVP are a common feature of the developing mouse hypothalamus. The functional role of this mixed neuronal population remains unexplored. However, it is tempting to speculate that OXT and AVP may share common signaling pathways during early development as immature forms of the OXT and AVP receptors may be activated by both neuropeptides during embryonic and early postnatal stages[55,56]. Furthermore, our results indicate that OXT and AVP neurons follow independent specification patterns in different hypothalamic nuclei. Together, these data show that OXT and AVP systems in the mouse brain are highly dynamic during early development and continue their remodeling until the postnatal period.

**Cell diversity among hypothalamic nuclei during development.** 3D analysis of OXT and AVP expression across the hypothalamus revealed a high degree of variability in their specification patterns. Thus most of the rostral nuclei, particularly ADPN, BNST, PeVN, and SCH, show higher degree of homogeneity (Fig. 6a–h) whereas caudal nuclei (SON, PVN, RCH, and AN) are

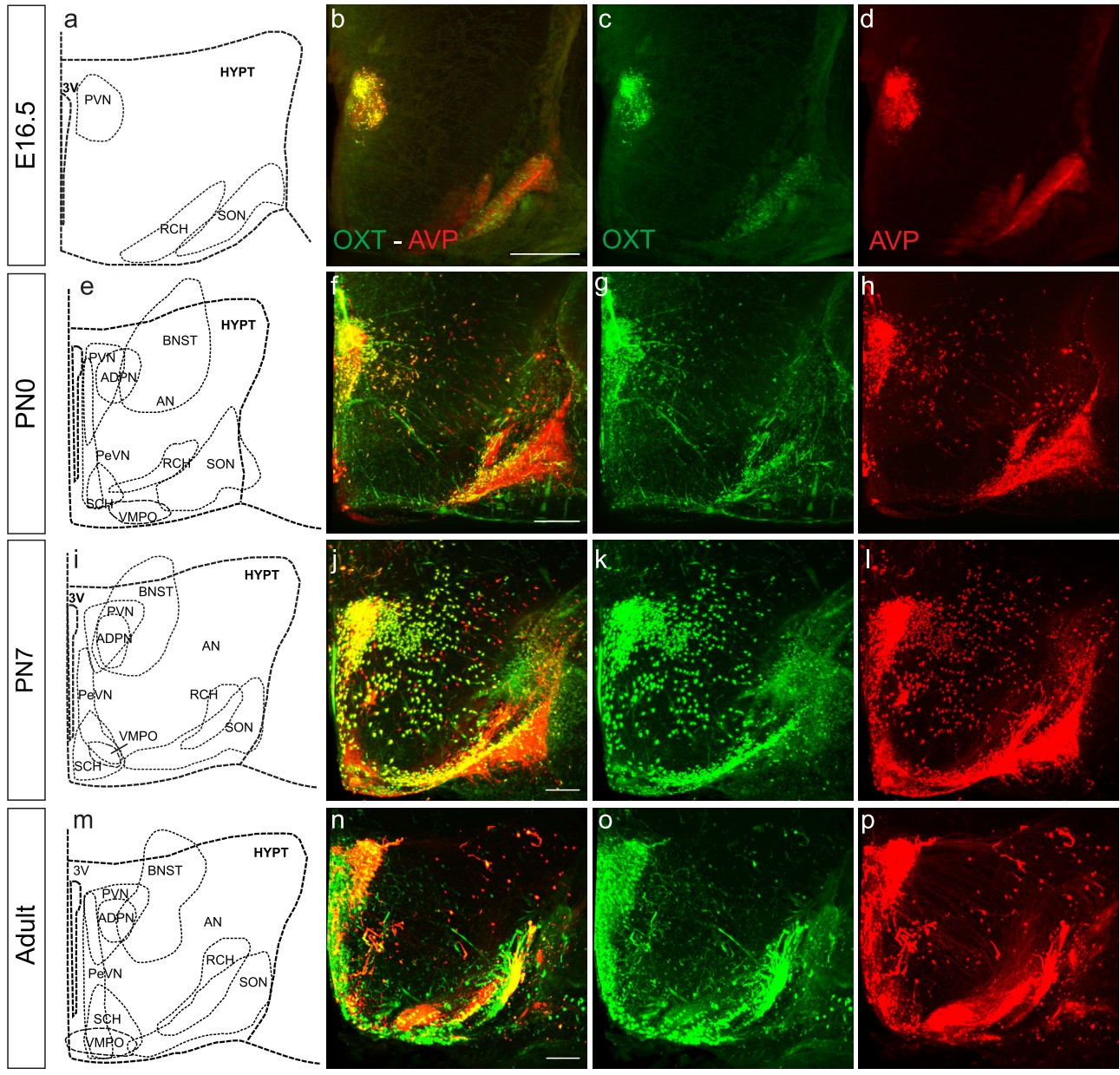

**Fig. 2 OXT and AVP expression patterns in the developing mouse hypothalamus.** Snapshot of a whole-brain stained with an anti-OXT (green) and anti-AVP (red) antibody. Brain coronal sections at multiple developmental stages: E16.5 (**a–d**), PN0 (**e–h**), PN7 (**i–l**) and young adult (**m–p**). 3v third ventricle, PeVN periventricular nucleus, ADPN anterodorsal preoptic nucleus, BNST bed nucleus of stria terminalis, VMPO ventromedial preoptic nucleus, SON supraoptic nucleus, PVN paraventricular nucleus, RCH retrochiasmatic area, AN accessory nucleus, SCH suprachiasmatic nucleus. Scale bar: 200 μm.

more heterogeneous containing a large number of OXT + AVP neurons during early development (Fig. 6i–p). Among the homogeneous areas, adult ADPN and BNST consist mainly of OXT neurons (Fig. 6ai–bi and e-f) whereas SCH is almost exclusively made up of AVP neurons[53] (Fig. 6di and h). Lastly, the PeVN exhibits an intermediate behavior with a low percentage of OXT + AVP neurons (Fig. 6ci–g). On the other hand, heterogeneous nuclei like the PVN can be subdivided in well-defined subregions where AVP neurons concentrate at the dorsolateral (rostral view, arrow in Fig. 6ii) and intermediate region (lateral view, arrow in Fig. 6m), similarly to what has been described in the rat brain[57]. Our results revealed that this distribution is maintained from newborn to adulthood. However, at early embryonic stages OXT + AVP neurons appear mostly located at the rostral PVN whereas AVP neurons concentrate in

the caudal area at E16.5. Similarly, OXT and AVP expression in the SON is restricted to defined areas, with a high percentage of OXT neurons in the rostral section and most AVP cells concentrated in the ventrolateral region[57] (Fig. 6ji and n). The RCH is also constituted by two well-differentiated areas of high cellular heterogeneity (Fig. 6ki and o), one along the surface (arrow in Fig. 6ki) and another surrounding the dorsomedial area (arrowhead in Fig. 6ki). Note these two RCH subregions appear highly interconnected (Supplementary movie 6), a feature that had not been appreciated previously, and that it seems a feature of early developmental stages. Among all nuclei, the AN exhibit the most pronounced cellular heterogeneity across all developmental stages because its rostral part is almost exclusively constituted by OXT and OXT + AVP neurons whereas the caudal region is formed mainly by AVP neurons (Fig. 6li and p).

**Table 2 Percentage of OXT, AVP, and OXT + AVP neurons in the hypothalamus nuclei during development.**

| OXT | E16.5 | PN0 | PN7 | Adult |
|---|---|---|---|---|
| BNST | 0.00 | 4.35 ± 4.35 | 7.418 ± 4.41 | 72.63 ± 24.28 |
| PVN | 10.65 ± 5.94 | 35.08 ± 7.33 | 19.54 ± 9.38 | 61.76 ± 4.29 |
| SON | 44.81 ± 3.16 | 15.79 ± 6.52 | 17.16 ± 9.96 | 45.06 ± 4.46 |
| VMPO | 0.00 | 34.29 ± 12.42 | 8.13 ± 8.13 | 41.55 ± 22.95 |
| RCH | 46.30 ± 5.70 | 30.91 ± 8.00 | 29.87 ± 7.97 | 43.84 ± 7.87 |
| ADPN | 0.00 | 0.00 | 18.40 ± 7.25 | 71.26 ± 23.78 |
| PeVN | 0.00 | 49.41 ± 15.75 | 17.90 ± 10.50 | 61.42 ± 20.64 |
| AN | 0.00 | 23.49 ± 9.16 | 4.35 ± 2.02 | 59.79 ± 1.68 |
| SCH | 0.00 | 0.69 ± 0.40 | 0.16 ± 0.16 | 2.22 ± 0.86 |
| MeA | 0.00 | 25.00 ± 25.00 | 0.00 | 25.00 ± 25.00 |

| AVP | E16.5 | PN0 | PN7 | Adult |
|---|---|---|---|---|
| BNST | 0.00 | 5.49 ± 4.02 | 4.90 ± 1.77 | 0.82 ± 0.33 |
| PVN | 28.23 ± 3.72 | 39.63 ± 7.15 | 12.30 ± 7.02 | 25.26 ± 6.12 |
| SON | 50.03 ± 2.70 | 58.93 ± 3.12 | 45.38 ± 2.49 | 34.45 ± 6.12 |
| VMPO | 0.00 | 22.80 ± 4.63 | 10.63 ± 7.10 | 8.10 ± 2.18 |
| RCH | 48.15 ± 5.82 | 38.04 ± 8.18 | 42.49 ± 4.61 | 32.74 ± 5.28 |
| ADPN | 0.00 | 11.46 ± 7.86 | 3.69 ± 1.90 | 2.24 ± 1.02 |
| PeVN | 0.00 | 21.90 ± 5.20 | 22.78 ± 8.09 | 7.55 ± 1.88 |
| AN | 0.00 | 57.53 ± 5.01 | 36.18 ± 6.04 | 34.62 ± 2.60 |
| SCH | 0.00 | 98.93 ± 0.48 | 99.08 ± 0.14 | 96.05 ± 1.22 |
| MeA | 0.00 | 0.00 | 22.69 ± 8.60 | 0.00 |

| OXT + AVP | E16.5 | PN0 | PN7 | Adult |
|---|---|---|---|---|
| BNST | 0.00 | 90.16 ± 8.34 | 87.68 ± 5.95 | 26.55 ± 24.07 |
| PVN | 61.12 ± 8.71 | 25.29 ± 14.44 | 68.16 ± 15.94 | 12.98 ± 4.06 |
| SON | 5.16 ± 0.93 | 25.28 ± 9.57 | 38.19 ± 9.90 | 20.48 ± 9.86 |
| VMPO | 0.00 | 42.92 ± 14.30 | 81.25 ± 11.25 | 50.35 ± 25.07 |
| RCH | 5.56 ± 0.75 | 31.06 ± 10.80 | 27.64 ± 3.78 | 23.43 ± 3.36 |
| ADPN | 0.00 | 88.54 ± 7.86 | 77.91 ± 8.45 | 26.50 ± 22.93 |
| PeVN | 0.00 | 28.69 ± 11.73 | 59.33 ± 18.03 | 31.03 ± 21.02 |
| AN | 0.00 | 18.98 ± 11.00 | 59.47 ± 6.30 | 5.59 ± 1.34 |
| SCH | 0.00 | 0.12 ± 0.12 | 0.54 ± 0.18 | 1.73 ± 0.79 |
| MeA | 0.00 | 25.00 ± 25.00 | 52.31 ± 17.91 | 50.00 ± 28.87 |

Quantification of percentage of each neuron population for different developmental stages (E16.5, PN0, PN7, and adult brain).
Results are expressed as mean ± S.E.M. (n = 4 brains).
BNST bed nucleus of stria terminalis, PVN paraventricular nucleus, SON supraoptic nucleus, VMPO ventromedial preoptic nucleus, RCH retrochiasmatic area, ADPN anterodorsal preoptic nucleus, PeVN periventricular nucleus, AN accessory nucleus, SCH suprachiasmatic nucleus, MeA medial amygdalar nucleus.

**Spatial distribution of OXT and AVP neurons in the developing PVN.** Spatial differences in the expression pattern of OXT and AVP neurons are likely due to distinct functional demands during development. We aimed to explore specific cell-type distribution dynamics analyzing 3D reconstructions of the PVN, a prominent brain source for OXT and AVP. Despite some subtle differences, OXT and AVP neurons follow a similar expression pattern in the midline and the rostro-caudal axis (Fig. 7). As such, during early developmental stages (E16.5, PN0), both OXT and AVP neurons can be found mostly distributed away from the midline to progressively populate the most medial regions as development progresses (Fig. 7a–h) but with some noticeable differences: whereas OXT neurons are more abundant in the region nearest to the midline, AVP neurons are concentrated in the intermediate regions (also see Fig. 6m and 6ii). This trend is also observed in the OXT + AVP neuron population (Fig. 7, 7i–l), which from a fairly homogeneous distribution in the proximal regions expand to the midline by PN7. In contrast, OXT and AVP expression exhibits different dynamics in the rostro-caudal axis. At early developmental stages, OXT and OXT + AVP neurons are abundant at the PVN caudal region whereas they ultimately appear uniformly expressed in the adult PVN (Fig. 7a–d and i–l). AVP neurons on the other hand are less abundant in the caudal axis and mostly concentrate in the intermediate regions (Fig. 7e–h). This diversity is likely to respond to different roles of the different hypothalamic nuclei as main sources of OXT and AVP innervation at distinct stages of development.

**Heterogeneity of oxytocinergic cells during early development.** In addition to wild type animals, we analyzed OXT expression in a commercial transgenic line. The OXT-Cre mice (Jackson Laboratories, ID 02423435) express a Cre recombinase sequence right after the stop codon of the OXT gene and they are an effective tool for identifying OXT neurons[47]. To this aim, we bred these animals with a tdTomato reporter line to identify OXT-expressing cells. An anti-RFP antibody was used to enhance the signal of OXT-tdTomato cells in combination with the anti-OXT antibody (PS38) used in iDISCO[+] experiments (Fig. 8). Analysis of the OXT-tdTomato mouse line over development revealed that OXT expression can be identified as early as E14.5 (Fig. 8a–aii) with no detectable signal prior, that stage (no OXT signal was detected at E12.5). In agreement with our previous results, OXT-tdTomato cells first appear in caudal nuclei like the PVN, SON, and RCH (Figs. 8, 2, and 3). Surprisingly, tdTomato-expressing cells do not perfectly colocalize with OXT positive neurons identified by immunohistofluorescence during embryonic and early postnatal stages. Remarkably just ~20% of the identified OXT neurons were RFP positive at E16.5 (22.81% ± 4.44; n = 3; Fig. 8b–bii). We further tested the ability of the PS38 anti-OXT

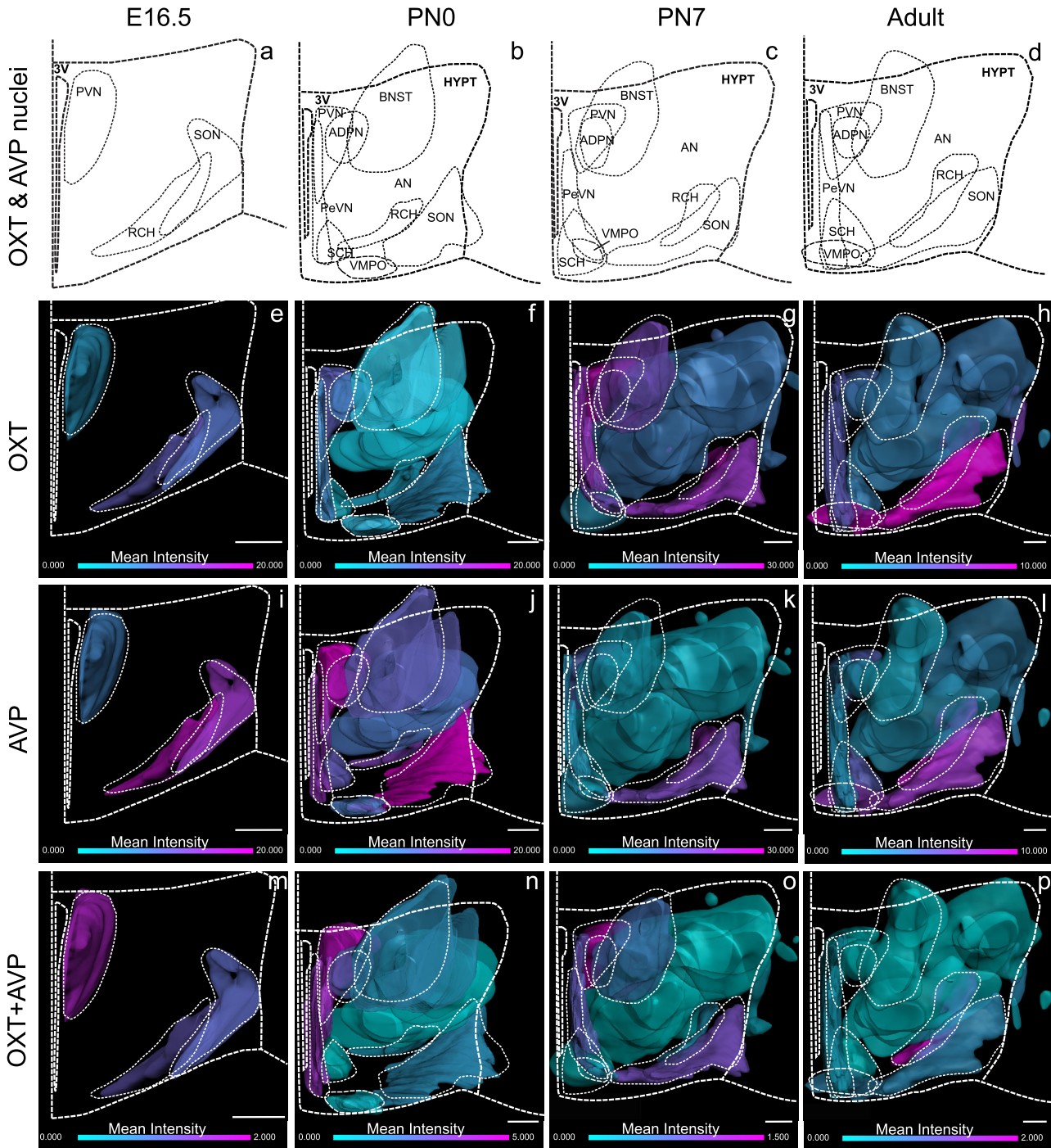

**Fig. 3 OXT, AVP, and OXT + AVP cell density in the developing mouse hypothalamus.** 3D hypothalamic flatmap representations of OTX (**e–h**), AVP (**i–l**) and OXT + AVP cells (**m–p**) in each nucleus at a range of developmental stages: E16.5 (**a**), PN0 (**b**), PN7 (**c**) and adult (**d**). Flatmaps show the expression intensity of each cell population at each developmental stage. Please note the difference intensity scale according to the developmental stage. The minimum intensity is represented in turquoise and the maximum in pink. 3v third ventricle, PeVN periventricular nucleus, ADPN anterodorsal preoptic nucleus, BNST bed nucleus of stria terminalis, VMPO ventromedial preoptic nucleus, SON supraoptic nucleus, PVN paraventricular nucleus, RCH retrochiasmatic area, AN accessory nucleus, SCH suprachiasmatic nucleus. Scale bar: 200 μm.

antibody to detect OXT expression at early developmental stages (Supplementary Figs. 3 and 4) comparing it with another commercial antibody (see "Material and Methods" for details) finding that PS38 was particularly suitable for early OXT identification (Figs. 1 and 2; Supplementary Figs. 3 and 4). Thus, the lack of colocalization between OXT and tdTomato in this mouse line, may suggest changes in the internal program of OXT positive cells likely due to functional changes during embryonic and early

postnatal development[58]. This heterogeneity is developmentally regulated as co-labeling increases as maturation progresses (Fig. 8c–f) to reach an almost complete co-localization in the adult PVN (Fig. 8f–fii) as previously described in this mouse line[47] (~92% co-expression in PVN and SON was reported in the original characterization by Wu et al.,[47]). However, these early OXT precursors persist until adulthood in particular areas like the SON and the RCH (Fig. 9). In fact, these nuclei retain a

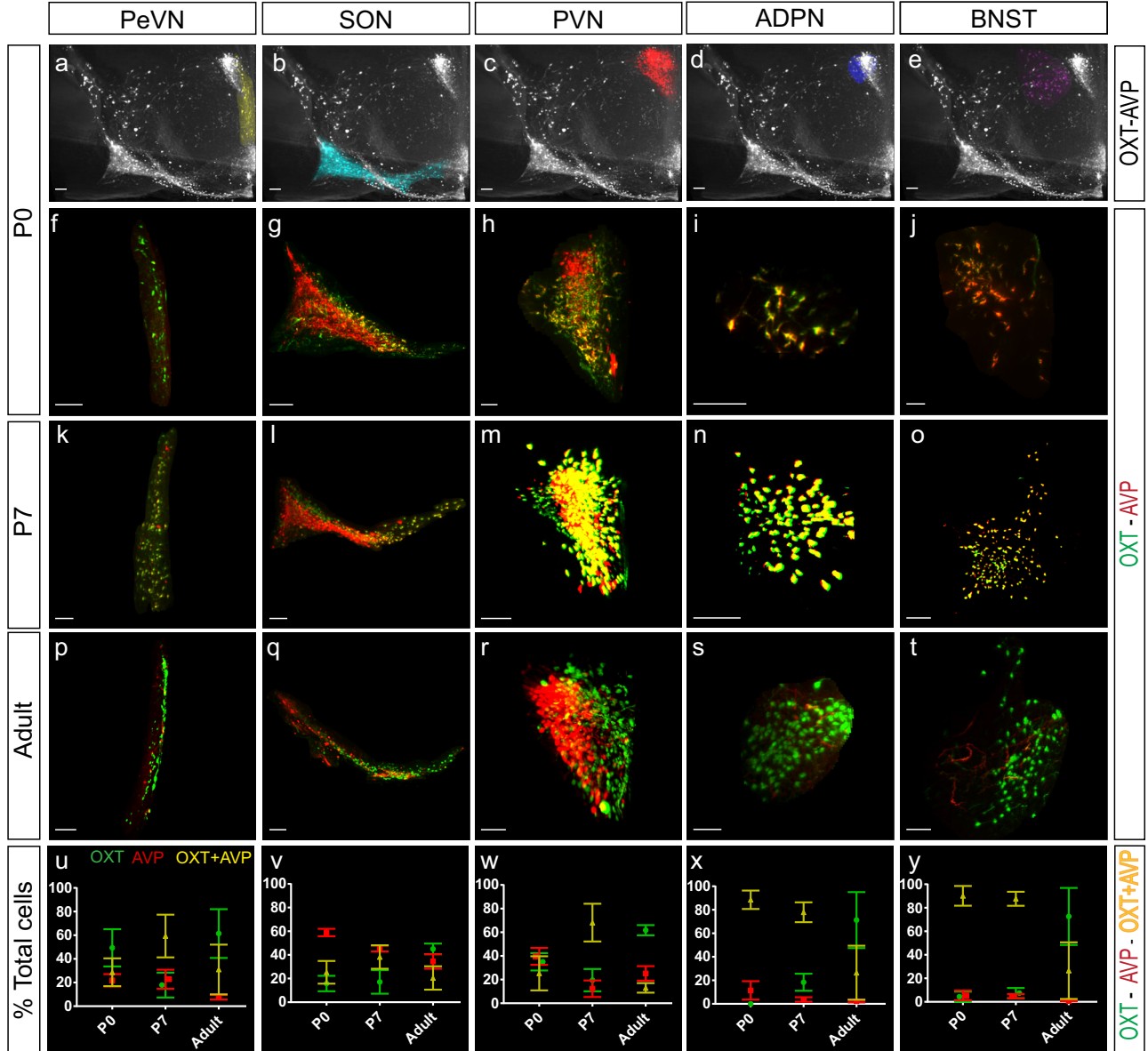

**Fig. 4 AVP and OXT systems in distinct hypothalamic nuclei differ in their temporal dynamics.** Whole-mount immunolabeling against OXT and AVP at PN0 mouse hypothalamus pseudocolored in each nucleus (**a–e**). Nucleus segmentation shows OXT (green) and AVP (red) at different stages: PN0 (**f–j**), PN7 (**k–o**), and young adult (**p–t**). Percentage of each neuronal population, OXT, AVP, and OXT + AVP neurons, in each nucleus during development (**u–y**; $n = 4$ brains per stage; data are represented as mean ± S.E.M). Abbreviations: PeVN periventricular nucleus, SON supraoptic nucleus, PVN paraventricular nucleus, ADPN anterodorsal preoptic nucleus, BNST bed nucleus of the stria terminalis. Scale bar: 100 μm.

significant number of this oxytocinergic subpopulation at both PN7 and in the adult brain, following a heterogeneous distribution in the RCH (Fig. 9a–ai and b–bi) and a highly organized pattern in the SON where these neurons form a distinctive lateral subregion (Fig. 9c–ci and d–di). These findings reveal a niche of molecularly distinct oxytocinergic neurons (not-recognized by PS38 but visible from E14.5 in OXT-tdTomato mice) that may retain some immature features in the adult SON and RCH.

## Discussion

Until now, a complete picture of the OXT and AVP innervation in the developing mouse brain has been lacking whereas more detailed information has been available for the rat brain. This lack of knowledge has prevented establishing meaningful correlations between anatomical and functional data obtained from transgenic

mouse lines and mouse behavioral studies[11,16–20]. Our work aims to overcome these current challenges by providing a detailed analysis of the OXT and AVP systems in the developing mouse brain. Here, we have implemented novel iDISCO+ clearing techniques to generate high cell-resolution brain-wide maps of the developing OXT and AVP systems. Systematic 3D cell quantification revealed that the formation of these circuits exhibits specific features distinct from the rat and other rodents which need to be considered when working with this animal model.

Similar to other rodents[58,59], mouse OXT and AVP are mostly synthesized in the PVN, SON, and AN. However, vasopressinergic and oxytocinergic cells can be found in other hypothalamic areas as well as non-hypothalamic regions, where these systems have been less studied. The cellular resolution achieved by the iDISCO+ clearing method revealed OXT and AVP neurons in non-hypothalamic areas like the MeA and particularly, the BNST

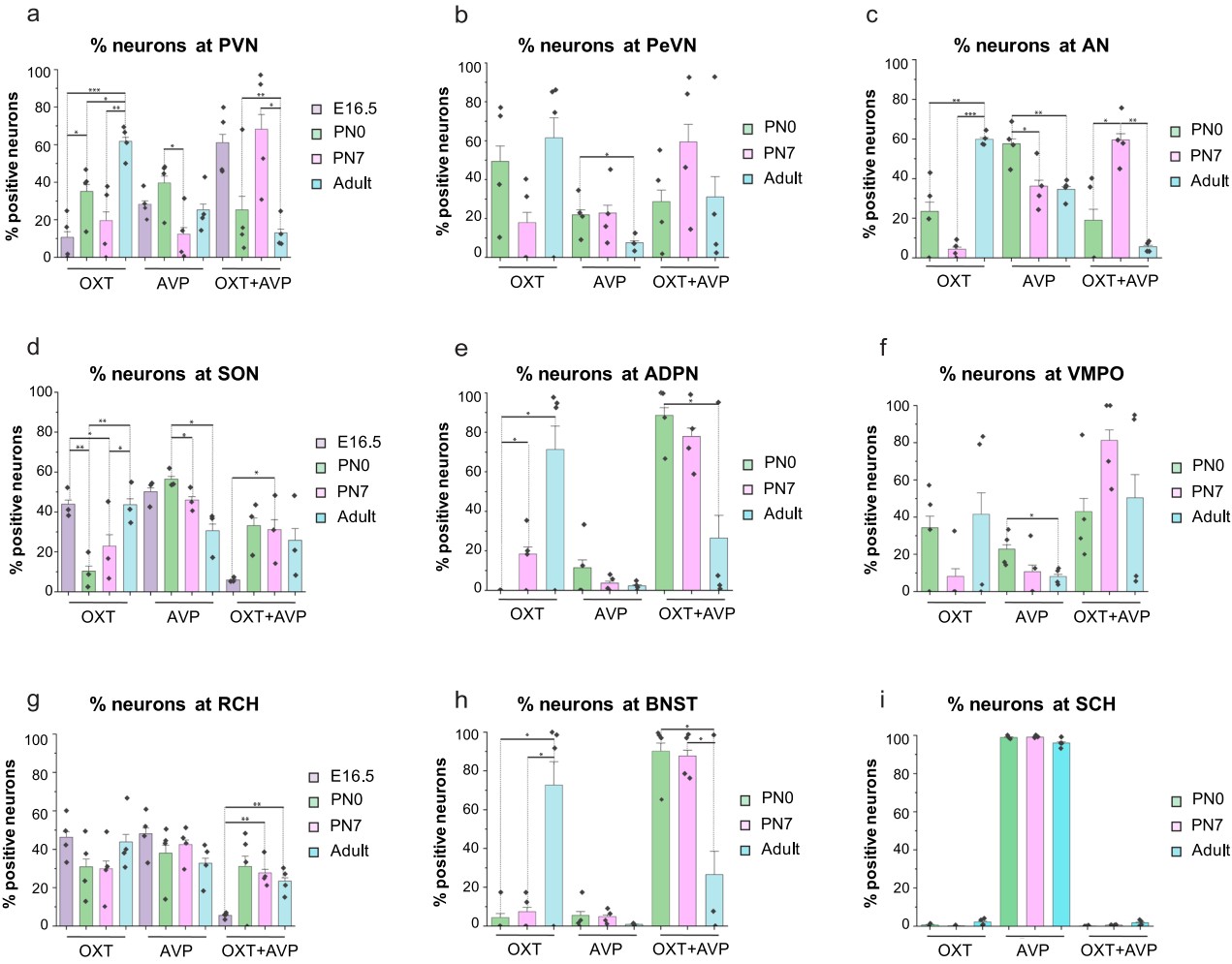

**Fig. 5 Quantification of OXT and AVP neurons during development.** Percentage of OXT, AVP and OXT + AVP neurons is represented at E16.5, PN0, PN7, and adult brain. OXT and AVP co-labeling peaks at PN7 and then decreases in favor of OXT expression in most nuclei with the exception of SCH which is primarily constituted by AVP-expressing neurons (**a–i**). Data are represented as mean ± S.E.M ($n = 4$) and data points are included on each graph. $T$-Student test, $p$-value % OXT: $PVN_{E16.5-PN0} = 0.041$; $PVN_{E16.5-Adult} = 0.00043$; $PVN_{PN0-Adult} = 0.02003$; $PVN_{PN7-Adult} = 0.00640$; $SON_{E16.5-PN0} = 0.00706$; $SON_{E16.5-PN7} = 0.03818$; $SON_{PN0-Adult} = 0.00999$; $SON_{PN7-Adult} = 0.04302$; $ADPN_{PN0-PN7} = 0.04423$; $ADPN_{PN0-Adult} = 0.02410$; $BNST_{PN0-Adult} = 0.03249$; $BNST_{PN7-Adult} = 0.03840$; $AN_{PN0-Adult} = 0.00800$; $AN_{PN7-Adult} = 0.000001$; $p$-value % AVP: $PVN_{PN0-PN7} = 0.03430$; $SON_{PN0-PN7} = 0.01465$; $SON_{PN0-Adult} = 0.01190$; $PeVN_{PN0-Adult} = 0.04089$; $VMPO_{PN0-Adult} = 0.02835$; $AN_{PN0-PN7} = 0.03459$; $AN_{PN0-Adult} = 0.00668$; $p$ – value % OXT + AVP: $PVN_{E16.5-Adult} = 0.00243$; $PVN_{PN7-Adult} = 0.01533$; $SON_{E16.5-PN7} = 0.01596$; $RCH_{E16.5-PN7} = 0.00122$; $RCH_{E16.5-Adult} = 0.00205$; $ADPN_{PN0-Adult} = 0.04295$; $BNST_{PN0-Adult} = 0.04666$; $BNST_{PN7-Adult} = 0.04869$; $AN_{PN0-PN7} = 0.01873$; $AN_{PN7-Adult} = 0.00016$. PeVN periventricular nucleus, ADPN anterodorsal preoptic nucleus, BNST bed nucleus of stria terminalis, VMPO ventromedial preoptic nucleus, SON supraoptic nucleus, PVN paraventricular nucleus, RCH retrochiasmatic area, AN accessory nucleus, SCH suprachiasmatic nucleus.

(Supplementary movies 1–5) where OXT- and AVP-expressing neurons have been difficult to identify by traditional approaches[36–39]. These findings are different from the rat brain where OXT and AVP neurons have not been described in the BNST at any developmental stage[40–46]. These differences in OXT and AVP expression may relate to distinctive functions of the OXT and AVP systems in the rat and mouse brain. As such, studies in OXT knock out mice indicated that, contrary to rats, OXT ablation does not prevent sexual or maternal behavior in mice despite being indispensable for the milk ejection reflex and lactation[4,60]. Interestingly, the mouse BNST has been suggested to participate in social recognition through an OXT-mediated signaling involving the MeA[61]. Thus the presence of a small population of OXT and AVP neurons in these areas in the mouse brain is suggestive of local peptidergic regulation which may be unique to mice. Furthermore, oxytocinergic and vasopressinergic systems appear to be sexually dimorphic in mice[62–69] even during early development[70], although sexual dimorphism may be may

less prominent in the developing AVP system[70,71]. Another interesting difference is the presence of a well-defined pool of AVP neurons in the dorsolateral SON in the mouse brain (Figs. 2, 4 and Supplementary movie 2), whereas AVP neurons in the rat have been reported to first appear in the SON ventromedial region[57].

In this study, we employed an anti-OXT antibody (PS38) that specifically recognizes OXT-neurophysin I[42,59]. The OXT gene is cleaved to generate multiple OXT intermediate forms and the carrier protein neurophysin I required to transport OXT to secretory granules[72]. Neurophysin I facilitates the transport of OXT throughout the secretory pathway and is released together with OXT, but its function as a signaling molecule remains unknown. Given that neurophysin I is required for OXT trafficking and storage, it is not surprising that the two proteins co-localize at the subcellular level[73] and that the expression pattern of the two is identical from embryonic stages to adulthood[74], indicating their maturation occurs in parallel and is

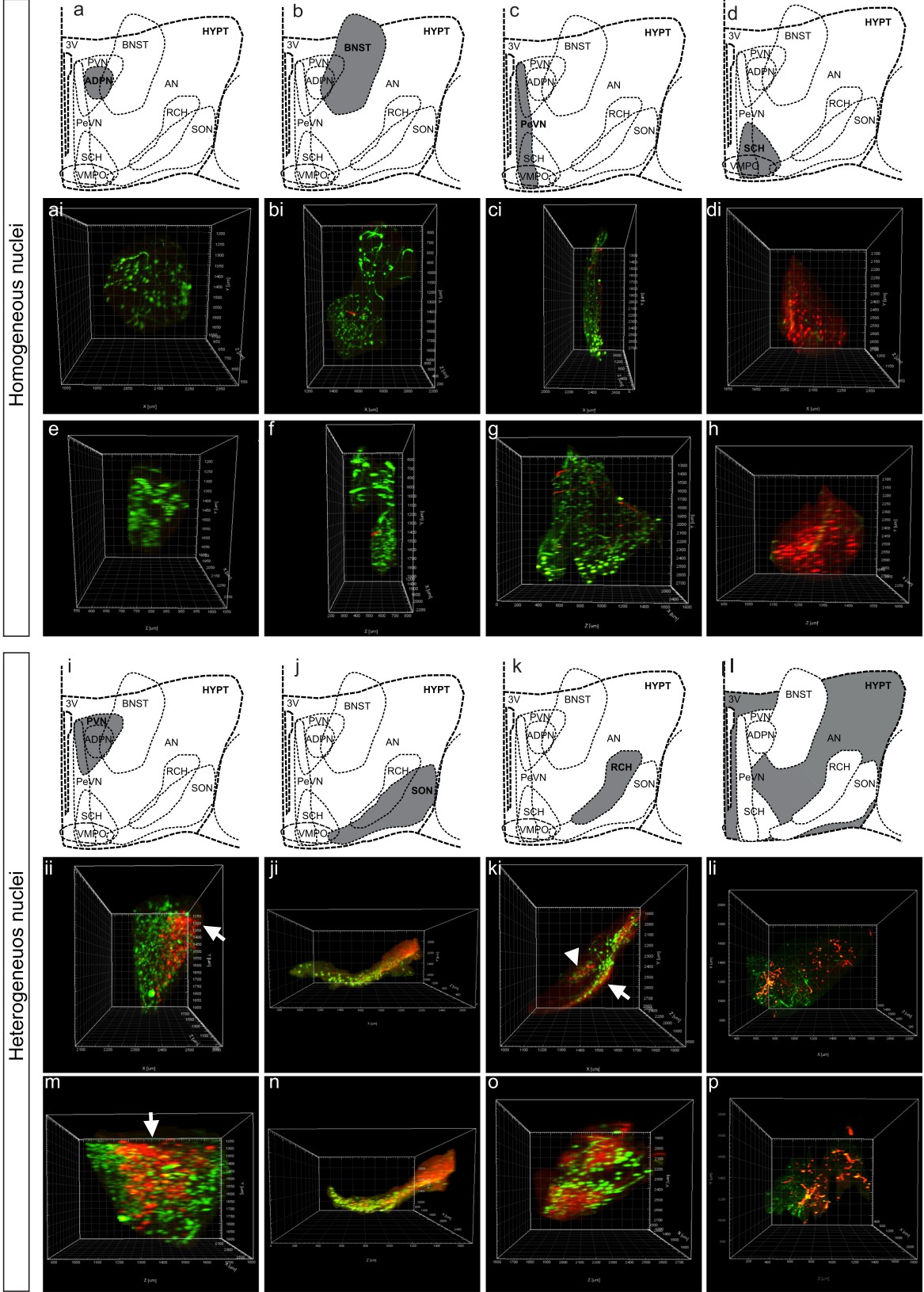

**Fig. 6 Cellular diversity within the hypothalamic nuclei.** Whole-mount immunolabeling against OXT and AVP in the adult mouse hypothalamus and surrounding areas such as the BNST. Nucleus segmentation shows OXT (green) and AVP (red) in distinct nuclei. The top panel shows the most homogeneous nuclei including ADPN, BNST, PeVN, and SCH (**a–h**) and the bottom panel the most heterogeneous ones, PVN, SON, RCH, and AN (**i–p**). Each nucleus (**a–d** and **i–l**) is represented by two images from different perspectives: a rostral (**ai–di** and **ii–li**) and a lateral view (**e–f** and **m–p**).

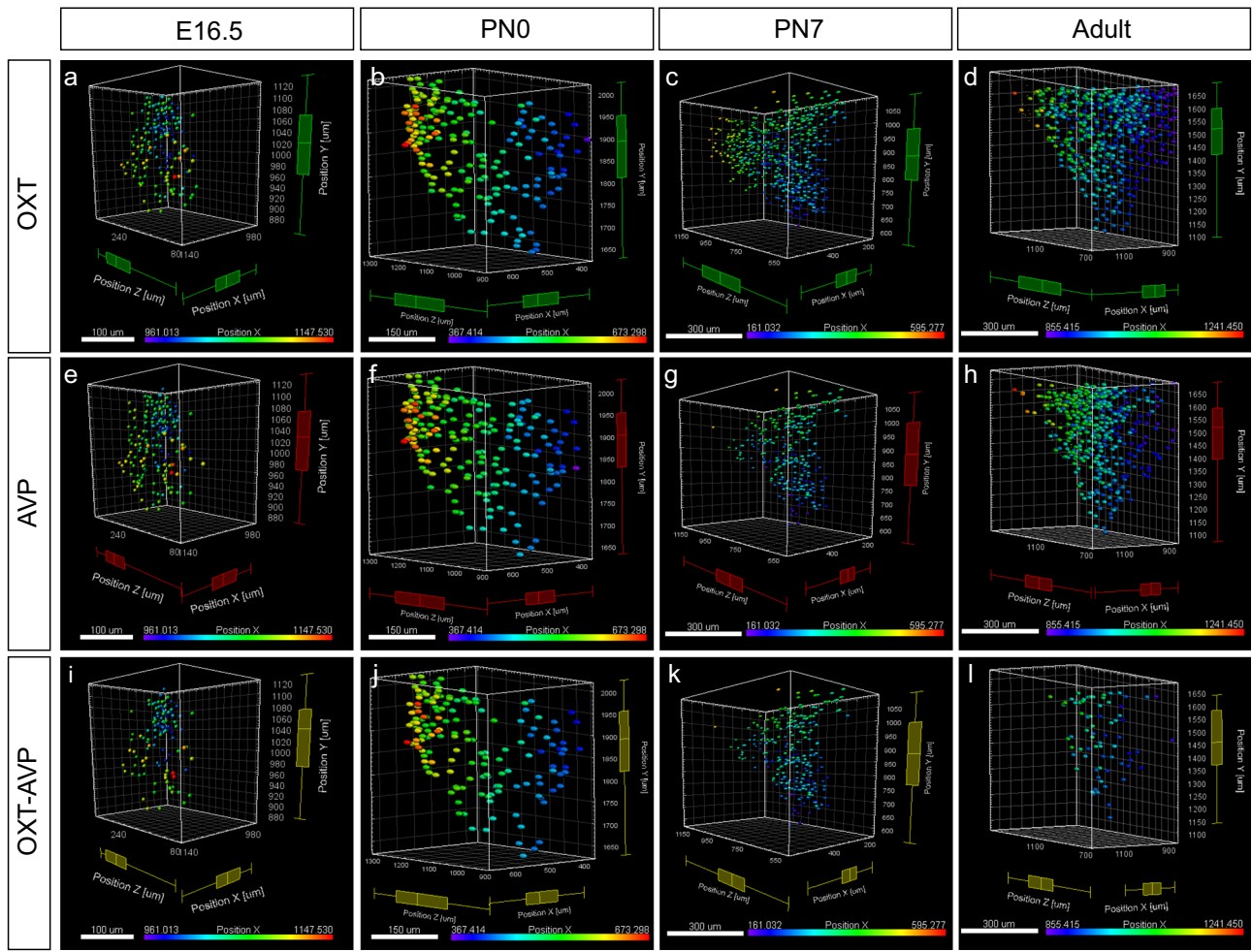

**Fig. 7 Spatial cell distribution in the PVN during development.** Cell distribution of OXT (**a–d**), AVP (**e–h**) and OXT + AVP (**i–l**) neurons are represented at E16.5, PN0, PN7 and adult brain. Each plot shows the cell distribution, represented as spots in the 3D brain reconstruction. Colors indicate distance to the midline. Each developmental stage is indicated with a histogram of different color. Each plot includes the minimum value, quartile 1 (Q1), median, quartile 3 (Q3) and the maximum value per each axis (x, y and z). *X position*: E16.5 OXT: min = 961.00; Q1 = 1009.00; median = 1046.00; Q3 = 1072.00; max = 1146.00; E16.5 AVP: min = 972.00; Q1 = 1014.00; median = 1050.00; Q3 = 1078.00; max = 1146.00; E16.5 OXT + AVP: min = 973.00; Q1 = 1006.00; median = 1040.00; Q3 = 1069.00; max = 1147.00; P0 OXT: min = 378.00; Q1 = 467.00; median = 526.00; Q3 = 569.00; max = 673.00; P0 AVP: min = 367.00; Q1 = 477.00; median = 525.00; Q3 = 561.00; max = 673.00; P0 OXT + AVP: min = 400.00; Q1 = 477.00; median = 529.00; Q3 = 568.00; max = 673.00; P7 OXT: min = 165.00; Q1 = 258.00; median = 313.00; Q3 = 371.00; max = 536.00; P7 AVP: min = 161.00; Q1 = 257.00; median = 309.00; Q3 = 357.00; max = 595.00; P7 OXT + AVP: min = 164.00; Q1 = 263.00; median = 302.00; Q3 = 346.00; max = 514.00; Adult OXT: min = 855.00; Q1 = 915.00; median = 958.00; Q3 = 1014.00; max = 1241.00; Adult AVP: min = 859.00; Q1 = 940.00; median = 981.00; Q3 = 1026.00; max = 1230.00; Adult OXT + AVP: min = 871.00; Q1 = 934.00; median = 971.00; Q3 = 1013.00; max = 1090.00. *Z position*: E16.5 OXT: min = 89.10; Q1 = 222.00; median = 270.00; Q3 = 307.00; max = 341.00; E16.5 AVP: min = 74.00; Q1 = 236.00; median = 273.00; Q3 = 304.00; max = 341.00; E16.5 OXT + AVP: min = 88.80; Q1 = 233.00; median = 268.00; Q3 = 308.00; max = 328.00; P0 OXT: min = 890.00; Q1 = 1043.00; median = 1143.00; Q3 = 1219.00; max = 1306.00; P0 AVP: min = 891.00; Q1 = 1042.00; median = 1128.00; Q3 = 1215.00; max = 1307.00; P0 OXT + AVP: min = 892.00; Q1 = 1039.00; median = 1147.00; Q3 = 1222.00; max = 1299.00; P7 OXT: min = 504.00; Q1 = 743.00; median = 871.00; Q3 = 1002.00; max = 1162.00; P7 AVP: min = 504.00; Q1 = 747.00; median = 825.00; Q3 = 929.00; max = 1153.00; P7 OXT + AVP: min = 501.00; Q1 = 723.00; median = 802.00; Q3 = 892.00; max = 1151.00; Adult OXT: min = 669.00; Q1 = 914.00; median = 1059.00; Q3 = 1245.00; max = 1471.00; Adult AVP: min = 678.00; Q1 = 991.00; median = 1091.00; Q3 = 1198.00; max = 1468.00; Adult OXT + AVP: min = 680.00; Q1 = 978.00; median = 1069.00; Q3 = 1168.00; max = 1329.00.

indistinguishable. This notion is further strengthen in our work where the PS38 antibody recognized OXT-expressing neurons from early developmental stages to adulthood in the mouse hypothalamus as reported previusly[40–46]. Interestingly, the expression of OXT and AVP neurons within distinct hypothalamic areas revealed distinct developmental dynamics. One of the most striking developmental patterns was observed in the SCH nucleus where the total number of AVP neurons declined significantly over time. Previous studies have reported an increase of hypothalamic volume in parallel with organism growth[75], thus

the decrease of the relative abundance of AVP neurons in the SCH likely reflects a switch in the internal program of these neurons rather than overall neuronal loss due to developmentally-regulated cellular apoptosis or pruning mechanisms. SCH is considered the master regulator of the circadian clock in mammals[76] which experiences drastic developmental changes as it transitions from non-photic cues, such as maternal hormonal cycles, to photic ones after birth[77,78], thus we hypothesize that the developmental dynamics of SCH AVP-expressing neurons is likely to reflect this functional adaption.

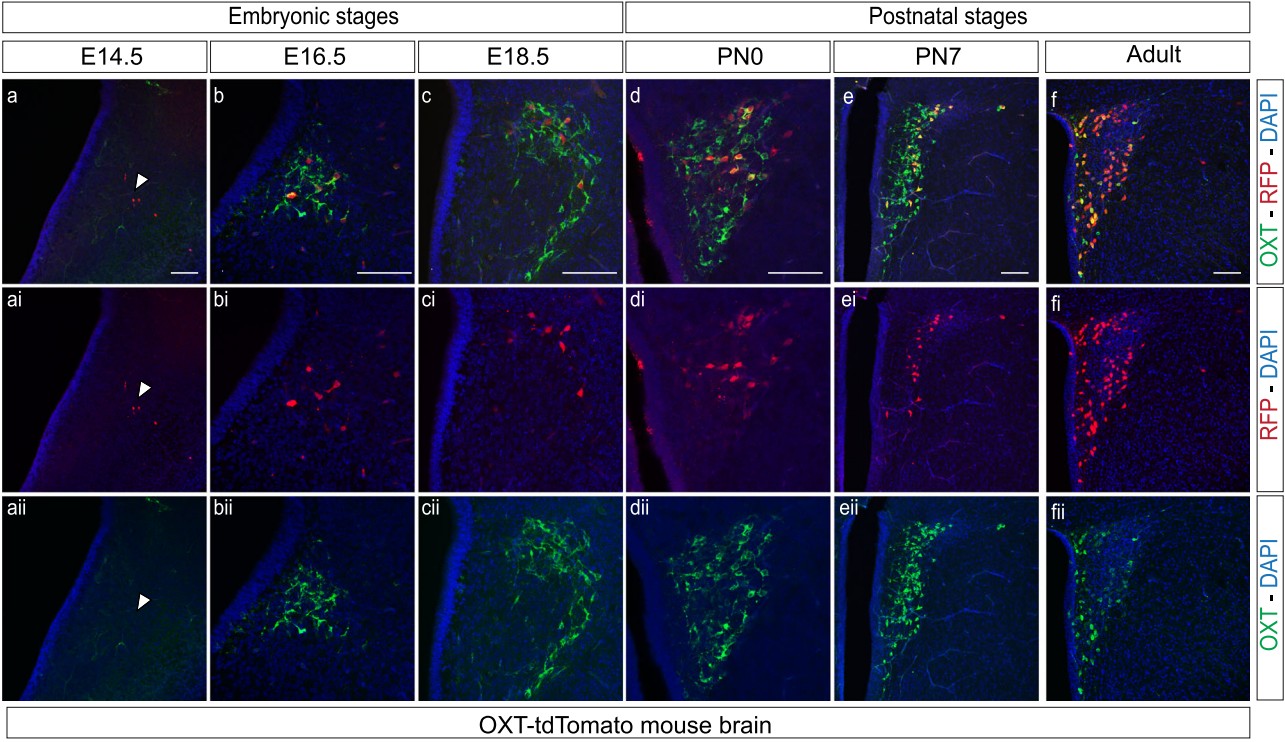

**Fig. 8 Analysis of OXT expression in an OXT-tdTomato mouse line.** Coronal sections of the PVN at a range of stages: E14.5 (**a–aii**), E16.5 (**b–bii**), E18.5 (**c–cii**), PN0 (**d–dii**), PN7 (**e–eii**) and adult (**f–fii**). Immunohistofluorescence against anti-RFP (red) and anti-OXT (green). Scale bar: 50 μm in **a–d**, 150 μm in **e**, **f**.

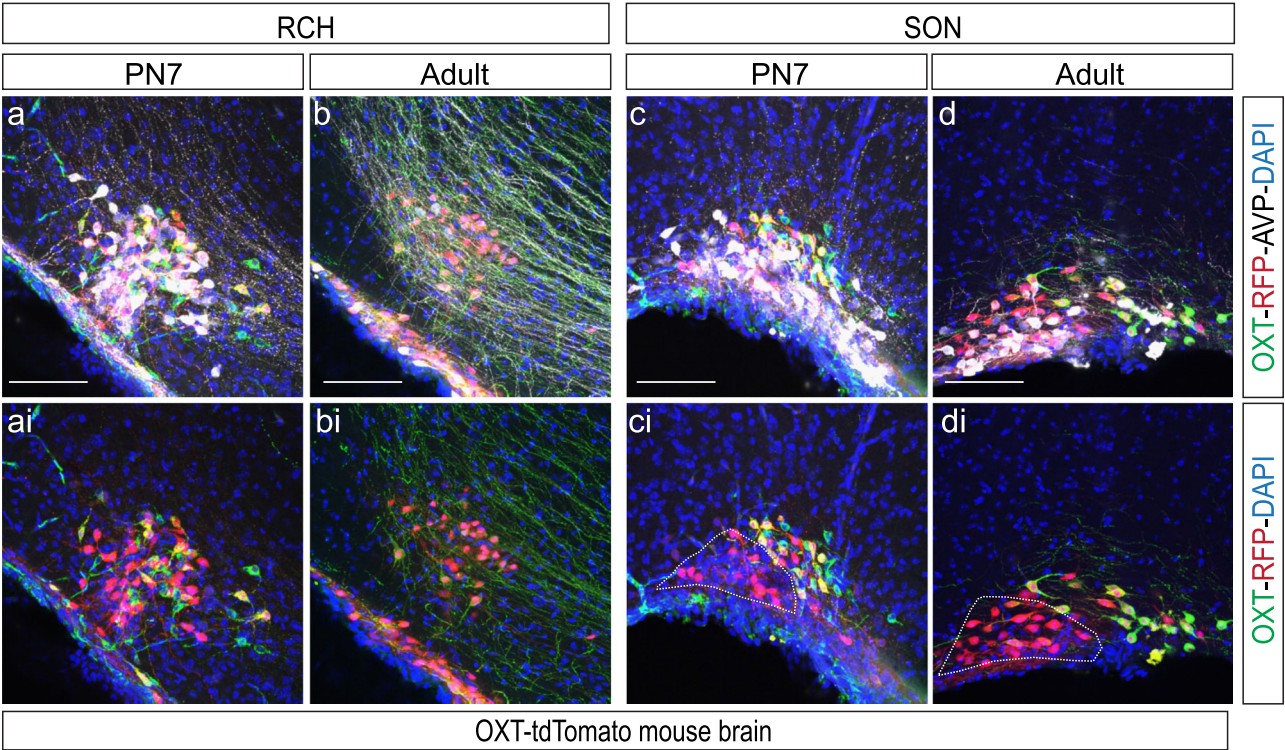

**Fig. 9 Development of SON and RCH in an OXT-tdTomato mouse line.** Brain coronal sections show the RCH (**a**, **ai**, **b**, **bi**) and SON (**c**, **ci**, **d**, **di**) at a range of stages: PN7 (**a**, **ai**, **c**, **ci**) including adult (**b**, **bi**, **d**, **di**). Immunohistofluorescence against anti-OXT (green), anti-RFP (red), and anti-AVP (white). White line represents the population of RFP positive cells not recognized by the PS38 OXT antibody. Abbreviations: SON supraoptic nucleus, RCH retrochiasmaticarea. Scale bar: 100 μm.

Parallel to the observed changes in SCH, our findings point to a substantial reorganization of the OXT/AVP balance in most hypothalamic nuclei during development (Figs. 2–4, and Tables 1 and 2). In general, our data indicate that the vasopressinergic system matures earlier than oxytocinergic neurons, which were found scarce at PN7 in agreement with previous reports[36–39]. Immunohistochemistry and in situ hybridization studies in the rat brain indicated the existence of several intermediate OXT forms during early development[59]. In this regard, the analysis of a transgenic OXT mouse line (OXT-tdTomato) revealed a significant percentage of oxytocinergic neurons that are not recognized by anti-OXT PS38 (Fig. 8) indicating the presence of distinct OXT precursors during early development. Given that the OXT-Cre transgenic line used in this study[47] was generated following a knock-in strategy that ensured the co-expression of OXT and Cre, discrepancies in OXT and RFP co-localization may be due to changes in the internal program of OXT positive cells likely due to functional changes during embryonic and early postnatal development[58,59]. Consisting with this notion, co-localization progressively increased over time suggesting that OXT processing is a developmentally regulated process[58,59]. However, OXT processing appears to exhibit region-specific properties since areas like the SON and RCH retain a significant number of OXT-tdTomato neurons (not recognized by the PS38 anti-OXT antibody) until adulthood (Fig. 9). Although the existence of immature non-amidated forms of OXT in the hypothalamus is well documented[58,59], to the best of our knowledge, this is the first report of an OXT precursor specific to the SON and RCH. To note, this OXT subpopulation is organized in a particular niche within the SON suggesting the existence of a defined area with potentially distinctive properties. Our studies indicate that the OXT-Cre line employed here will be a useful tool to carry out functional analysis of this particular subpopulation of OXT neurons.

Little is known about the function of OXT and AVP peptides during early development. However, immature OXT forms coexist with the mature amidated form during the entire postnatal life[58] suggesting that these immature precursors may contribute to hypothalamic neuromodulation by means that remain to be elucidated. OXT seems to undergo a more intensive maturation program than AVP, which has fewer immature precursors and a mature form that can be detected as early as E16.5[43]. In contrast, amidated OXT is first detected at birth and subsequently coexists with immature precursors until late postnatal stages[36,46]. This phenomenon is well-documented in prairie voles[79], a trend observed in the present study and suspected to be common in other mammals including humans. Intriguingly, some immature OXT forms are capable of activating the oxytocin receptor, which is widely expressed before birth[80], suggesting a role of these precursors in the consolidation of the oxytocinergic circuit. In fact, magnocellular OXT neurons exhibit undeveloped morphology and electrophysiological properties at birth that are progressively refined by reducing spontaneous activity and decreasing action potential-evoked calcium entry during the first postnatal weeks[81–83]. These changes are accompanied by a switch in the regulatory mechanisms of cytosolic calcium from extrusion to sequestration into the endoplasmic reticulum[84] which has been postulated to impact neuropeptide exocytosis[84]. In this scenario, the appearance of amidated OXT may influence the magnocellular neuron maturation as OXT modulates neuronal excitability[85], contributing to shape synaptic transmission and plasticity. Consistent with this notion, our analysis has revealed that the AVP and OXT circuits undergo two periods of intense remodeling. The first occurs at time of birth, and coincides with a critical period for social behavior. The second occurs at PN7, when the OXT/AVP ratio appears to increase in most hypothalamic nuclei. The functional consequences of this rearrangement are unknown, but these adjustments may underlie a critical period for hypothalamic maturation during the first two postnatal weeks[81–83], which is likely involve profound cellular plasticity events as neurotransmitter switches[86]. This complex scenario is further expanded in the present study by the finding of tissue-specific OXT precursors (Fig. 9) in the SON and RCH which are likely to differentially impact the maturation of the oxytocinergic system in these particular regions and their target sites. Furthermore, OXT has been reported to modulate astroglial and microglial function[87,88] by activating mechanisms that impact brain homeostasis and immunoresponse, as well as social behavior[87,88].

In summary, the present study highlights the importance of applying novel imaging tools to analyzing the properties of specific circuits in the whole brain. Improved brain clearing methods in combination with light sheet fluorescent microscopy and 3D reconstruction allow high resolution analysis of specific pathways to address their dynamics during physiological and pathological conditions. The potency of these novel technologies has revealed the developing mouse hypothalamus as a dynamic structure that undergoes intense remodeling up to the postnatal period with different nucleus exhibiting distinct temporal and spatial dynamics of the specification of OXT and AVP pathways. Our findings provide new information to understand the specification of these neuronal circuits, which is a critical step for uncovering the wiring alterations and cellular dysfunctions underlying social deficits.

## Methods

**Animals**. Experiments were performed in wild type ICR mice bred in-house. Brains were extracted at E16.5, PN0, and PN7 from young female adults (2–3 months). Data from female and male animals were pooled for E16.5, PN0, and PN7 animals. A limitation of the study is the use of a mixture of male and female brains to analyze early developmental stages (E16.5, PN0, and PN7) hindering the detection of potential sex differences. Oxytocin-Ires-Cre:Rosa26iDTR/+OXT-Cre (OXT-Cre thereafter) mice were obtained from Jackson Laboratories (strain ID 024234). These animals express Cre recombinase under the control of the endogenous OXT promoter and has been used previously to specifically target oxytocinergic neurons[47]. In this configuration, Cre and Oxt are transcribed as a single bicistronic mRNA and the IRES sequence allows simultaneous translation of both OXT and Cre proteins from the same transcript[47]. OXT-Cre mice were bred with a tdTomato reporter line (Ai9, Jackson Laboratories strain ID 007909) which exhibit a Cre reporter allele with a loxP-flanked STOP cassette preventing transcription of the red fluorescent protein variant tdTomato inserted into the Gt(ROSA)26Sor locus. OXT-tdTomato mice express robust tdTomato fluorescence following Cre-mediated recombination. All experiments were performed according to Spanish and European Union regulations regarding animal research (2010/63/EU), and the experimental procedures were approved by the Bioethical Committee at the Instituto de Neurociencias and the Consejo Superior de Investigaciones Científicas (CSIC). Animals were housed in ventilated cages in a standard pathogen-free facility, with free access to food and water on a 12 h light/dark cycle.

**Immunohistofluorescence**. Mice were intracardially perfused and brains were extracted and fixed in 4% paraformaldehyde (PFA) in PBS overnight. Embryos were fixed overnight by immersion in 4% PFA. Samples were embedded in agarose (4%) and sectioned at 50 μm using a Leica VT1000 S vibratome. Sections were rinsed three times in PBS and incubated 1 h in PBST (0.3% Triton, 2% Normal Goat Serum). Samples were incubated overnight at 4 °C in a rotating shaker with the following primary antibodies: mouse anti-oxytoxin PS38[42,59] (1:800; kindly provided by Dr. Harold Gainer, NIH); rabbit anti-oxytocin (1:800; Millipore, AB911); rabbit anti-RFP (1:1000; Abcam, ab62341); rat anti-RFP (1:1000; Chromotek, 5f8-100); rabbit anti-vasopressin (1:800; Millipore, PC234L). Note that mouse anti-OXT (PS38) and rabbit anti-OXT label a highly overlapping pool of OXT neurons in the adult brain (Supplementary Fig. 4), but rabbit anti-OXT is not able to identify OXT neurons at early developmental stages (Supplementary Fig. 5), in contrast to PS38 which detects OXT-positive cells as early as E16.5 (Figs. 2 and 3). AVP antibody recognizes the mature peptide which is detected as early as E16.5 indicating that AVP maturation occurs much earlier than OXT maturation. Tissue was rinsed three times in PBS and incubated with the corresponding secondary antibody: goat anti-mouse Alexa-488 (1:500; Invitrogen, A32723); donkey anti-mouse Alexa-647 (1:500; Jackson ImmunoResearch, 715-605-150); goat anti-rabbit Alexa-594 (1:500; Invitrogen, A11072); donkey anti-rat Alexa-594 (1:500; Jackson

ImmunoResearch, 712-585-153). After rinsing sections were incubated 10 min at room temperature (RT) with 0.001% DAPI (4',6-diamidino-2-phenylindole dihydrochloride, Sigma, D9542) in PBS for nuclear staining. Slices were mounted with Mowiol 40-88 (Millipore, 475904) for histology.

**Immunolabeling-enabled three-dimensional imaging of solvent-cleared organ (iDISCO+).** Whole-mount immunostaining and iDISCO+ optical clearing was performed as described in Renier et al.,[50] and Renier et al.,[51] Mice were intracardially perfused and brains were fixed in 4% PFA in PBS overnight. Embryos were fixed in 4% PFA immersion overnight. Brains were dehydrated using a series of concentrations of methanol (50, 80, 100, and 100%) and then incubated in 6% H$_2$O$_2$ in methanol overnight in order to bleach the samples. After rehydratation, samples were blocked with PBS-GT (0.5% Triton X-100, 0.2% gelatin) for durations that depended on developmental stage: overnight for E16.5 brains; 2 days for PN0 and PN7 brains; and 4 days for adult brains. Brains were incubated with OXT and AVP primary antibodies (mouse anti-oxytocin, Gainer Lab, NIH, PS38[42,59] (1:1000) and rabbit anti-vasopressin, Millipore PC234L (1:2000)) diluted in 0.1% saponin (Sigma, S4521) PBS-GT solution for 5 days (E16.5), 10 days (PN0), 2 weeks (PN7) or 3 weeks (adult brain) at 37 °C on a rotating shaker. Afterward, samples were incubated with secondary antibodies directly conjugated to a fluorophore donkey anti-mouse Alexa-647 (1:500; Jackson ImmunoResearch, 715-605-150) or goat anti-rabbit Alexa-594 (1:500; Invitrogen, A11072) at 37 °C overnight or two days for adult brains. Immunostaining controls using secondary antibodies (without primary antibodies) were performed to test the specificity of the primary antibodies for OXT and AVP.

For the clearing, brains were dehydrated in 20, 40, 60, and 80% methanol in H$_2$O at RT in a rotating shaker. Shaker incubations were for 1 h for embryonic and postnatal brains, and 1.5 h for adults. Specimens were then incubated twice in 100% methanol for 1–1.5 h, and treated overnight in 1 volume of 100% methanol and l/3 volumes of 100% dichloromethane (DCM; Sigma-Aldrich; 270997). On the next day, brains were incubated in 100% DCM for 30 min. Lastly, samples were cleared in 100% dibenzylether (DBE; Sigma-Aldrich; 108014) until becoming translucent.

**Microscopy and imaging processing.** Confocal microscopy was performed with a laser scanning Leica SPE-II DM5550 microscope with ×10, ×20, and ×40 objectives. Brains were imaged on a bidirectional triple light-sheet microscope (Ultramicroscope II, LaVision Biotec) equipped with a binocular stereomicroscope (MXV10, Olympus) with a ×2 objective (MVPLAPO, Olympus) using a 5.7 mm working distance dipping cap. Samples were placed in an imaging reservoir made of 100 % quartz (LaVision BioTec) filled with ethyl cinnamate (Aldrich, 112372-100G). The step size in Z-orientation between each image was fixed at 3 μm for 2.5× and 4× magnifications for nuclei identification and cell counting, and 5 μm for 0.63×, 1×, and 1.6× magnifications for whole-brain images analysis. 3D imaging was performed using ImspectorPro software (LaVision BioTec) and processed by the Imaris x64 software (version 9.2.1, Bitplane). Video analysis was optimized for cell identification.

**Nuclei identification, cell counting, and positioning.** Quantifications of OXT and AVP cells in each hypothalamic nucleus were calculated using alternate hemispheres of four brains. Brains at different developmental stages were processed and quantified using Imaris x64 software (version 9.2.1, Bitplane) using the tools for 3D reconstruction and segmentation. Boundaries of hypothalamic nuclei were identified according to the mouse brain atlas of Franklin and Paxinos[89]. The volume estimation of each nucleus was generated using immunohistochemical labeling as a guide and the Surface Imaris tool. False colors were assigned to these brain areas with the Surface Imaris tool. High-quality 3D images were selected for cell quantification. For accurate quantification of AVP- and OXT-expressing cells the fluorescence threshold was set manually and false positives were discarded from the final analysis using Imaris Spots tool. It is notable that at embryonic stages the image resolution, is not as sharp due to a greater abundance of blood vessels and neuronal processes. 3D pictures and movies were generated with Imaris. Cell 3D positioning was achieved by the statistical data-visualization module of Imaris, vantage view. High dimensional color plots were generated to analyze the 3D distribution of each cell population in the three axes (x, y and z).

**Statistics and reproducibility.** Several rounds of experiments were performed to assure the accuracy of the results. Samples were processed simultaneously and measurements were performed once the staining protocols were completed in no particular order. The authors were not blind to the age of animals during the processing of the samples since the size of the brain correlates with the age. However, the authors were blind during the analysis. Data were analyzed with Imaris and exported to Excel (Microsoft, Inc.) for data handling. Sample size were estimated using the power analysis set to significance level <0.05 and 0.80 to achieve over 80% of power with n = 4 samples per group. For statistical analysis, cell counts were estimated to follow a negative binomial distribution and adjusted to account for false positives. Samples were discarded when exhibiting poor immunostaining or deficient clearing. Results in the bar graph plot are expressed as the mean ± S.E.M. Statistically significant differences were determined using an unpaired t-test. $P < 0.05$ was considered statistically significant.

**Reporting summary.** Further information on research design is available in the Nature Research Reporting Summary linked to this article.

## Data availability
All the data are available from the corresponding authors upon reasonable request. There are no restrictions on the data availability.

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

## Acknowledgements

This work was supported by grants of the Spanish Ministry of Science and Innovation under the grant SAF2017-82524-R (to S.J.), the "Severo Ochoa" Program for Centres of Excellence in R&D (SEV-2013-0317 and SEV-2017-0723) and the Generalitat Valenciana through the program Prometeo/2019/014 (to S.J.). The mouse anti-OXT antibody was kindly provided by Dr. Harold Gainer, NIH. We thank Dr. Cristina Marquez (IN) for her insightful comments and Dr. Juan Antonio Moreno Bravo for his help with video editing and technical issues. We also are grateful to all Jurado Lab members for their support during the realization of this work and their assistance editing the manuscript.

## Author contributions

The project was led by S.J. Brain sample preparation, data acquisition, and analysis was done by M.P.M. Result interpretation, manuscript preparation, and editing was done by S.J. and M.P.M. with help of members from the Jurado Lab and the Instituto de Neurociencias (CSIC-UMH).

## Competing interests

The authors declare no competing interests.
