## [Peer Review File · Communications Biology]

Reviewers' comments:

Reviewer #1 (Remarks to the Author):

This manuscript presents beautifully the distribution of oxytocin-neurophysin (Oxt) and vasopressin peptide (Avp) in the developing mouse brain using the iDISCO+ protocol. I think this will be an important contribution to the field after the authors address some concerns.

1. The authors need to have a reader fluent in English go over the manuscript. Grammatical errors are too numerous, even in the introduction, and distract from the paper.
2. I have a hard time understanding how the authors can reach conclusions about the relative onsets of expression of the two nonapeptides when the Avp antibody is generated against the peptide whereas the Oxt antibody (PS38) is directed against the Oxt-specific neurophysin protein. What is known about how the Oxt peptide and its neurophysin correlate in their developmentally detected expressions. Further, whereas the Avp antibody may be specific for the processed neuropeptide (is this known for this Millipore antibody?), the PS38 Oxt-neurophysin-specific antibody says nothing about Oxt maturation.
3. Further, their statement beginning at line 160 is misleading as the authors are, in fact, not looking at the small protein-like (what does this mean?) molecule Oxt, but its co-expressed protein neurophysin which is much larger.
4. I am not convinced by the data that the Oxt-Cre line generated td-Tomato expression is necessarily indicative of natural Oxt gene expression. I would need to see co-expression of Cre (or td-Tomato) and Oxt transcripts. A possibility is that the Oxt-Cre line ectopically expresses Cre, at least temporally, out of synch from the Oxt locus.
5. In figure 3, please change the figure label to "Oxt+Avp", not "Oxt-Avp." At first I thought there was a distribution difference being displayed.

Reviewer #2 (Remarks to the Author):

Overall, this is an interesting paper that has utilized a state-of-the-art approach to evaluate the developmental timeline of oxytocin and vasopressin expression during development. Specifically, by using iDISCO+ coupled with light sheet microscopy the authors have provided a look into the development of these systems that has not been achieved before. There is no doubt that this work makes an important contribution to the field. However, as written, the paper oversimplifies these systems. Also, generally speaking, the manuscript seems under cited. For example, it is unlikely that oxytocin and vasopressin "underlie neuropsychiatric disorders", this is an overstatement. Most often, these systems are dysregulated and/or affected by drug treatment in some individuals with specific neuropsychiatric disorders. There is almost no mention of sex differences in these systems, which is critically important given that there are identified sex differences in their development, as well as in adult animals, that are thought to be associated with their function. While the authors indicate that adult female brains were used, the embryonic and early postnatal timepoints make no mention of the sex of the embryos/newborn pups used. This is relevant because, for instance, Tamborski et al (2016) found no oxytocin mRNA to be present at E16.5 in the male mouse brain. Lastly, how do the authors know that the oxytocin and vasopressin are expressed in neurons at E16.5 if no neuronal marker was utilized?

Minor issues:

- Figure 1A is fuzzy.
- The nomenclature for AVP is not maintained in the supplemental.

We are pleased to submit a revised version of our manuscript entitled "*Specification of oxytocinergic and vasopressinergic circuits in the developing mouse brain*". We highly appreciate reviewers' positive comments and suggestions, and as you will see we have addressed all their points in the best of our ability.

Please find attached the detailed response to reviewer's comments.

Reviewer #1

1.- *The authors need to have a reader fluent in **English** go over the manuscript. Grammatical errors are too numerous, even in the introduction, and distract from the paper.*

We apologize for the grammatical errors in the previous version, and highly appreciate reviewer's suggestion. In fact, we realize that some of the questions regarding the interpretation of our findings may have been due to poorly constructed sentences. The current manuscript has been thoroughly revised by a native speaker. The new version is easier to read and we do hope it meets the expectations of the reviewers and the editorial team.

2.- *I have a hard time understanding how the authors can reach conclusions about the relative onsets of expression of the two nonapeptides when the Avp antibody is generated against the peptide whereas the Oxt antibody (PS38) is directed against the Oxt-specific neurophysin protein. **What is known about how the Oxt peptide and its neurophysin correlate in their developmentally detected expressions.** Further, whereas the Avp antibody may be specific for the processed neuropeptide (**is this known for this Millipore antibody?**), the PS38 Oxt-neurophysin-specific antibody says nothing about Oxt maturation.*

We appreciate reviewer's comment which highlighted the need of including more information about the processing and maturation of the OXT peptide. To clarify this issue, we have now expanded the discussion (*changes in the manuscript are indicated in red*, lines 323-332) and the Material and Methods section (lines 426-430) with more details about the antibodies used in this study. We have further explored the ability of mouse anti-OXT PS38 to detect OXT at different maturation stages. These results are now shown in two new Supplementary Figures (new **Supplementary Figures 3 and 4**).

As indicated by the reviewer, the oxytocin gene encodes the pre-pro-OXT-neurophysin I peptide (**Fig. 1**, adapted from Grinevich et al., 2015) which is cleaved to generate different OXT intermediate forms and the carrier protein neurophysin I, required to transport OXT to secretory granules (Rao et al., 1992). Neurophysin I facilitates the transport of OXT throughout the secretory pathway and is released together with OXT (Stern et al., 1986; Mason et al., 1986), although its function as a signalling molecule remains unknown. Given that neurophysin I is required for OXT trafficking and storage, it is not surprising that the two proteins co-localize at the subcellular level (Sofroniew et al., 1979; Rhodes et al., 1981; Belenky et al. 1992) and that the expression pattern of the two is identical from embryonic stages to adulthood (Butovsky et al. 2006), indicating their maturation occurs in parallel and is indistinguishable.

Fig. 1. Biosynthesis of the different forms of OXT. This scheme represents the genomic structure of the oxytocin gene, which is contiguous to the *AVP* gene. The transcription and translation steps of the OXT gene are then indicated. First, OXT pre-pro-hormone is produced, and then cleaved by successive enzymes. OXT intermediate forms are already synthesized at E16.0 but the mature amidated OXT form is only detected at birth. *Figure adapted from Grinevich et al., 2015.*

Our study employs a monoclonal mouse anti-OXT (PS38) synthesized in Dr. Harold Gainer's laboratory. This antibody recognizes OXT-specific neurophysin I with high specificity (Ben-Barak et al., 1985; Whitnall et al., 1985), and since its generation has been widely-used for the study of the oxytocinergic system (Kogami et al., 2020; Hasan et al., 2019; Althammer et al., 2017; Otero-García et al., 2016; Grinevich et al., 2015 among many others). Previous neuroanatomical studies using PS38 showed that this antibody identifies the expression of OXT at E14.5 (**Fig 2B**, adapted from Grinevich et al., 2015). An antibody against the OXT intermediate VA10 revealed the appearance of this OXT form at E16.5 in a non-overlapping manner with PS38 signal (**Fig 2C**, adapted from Grinevich et al., 2015) suggesting that PS38 may recognize a different OXT variant. However, immunodetection with PS38 and VA10 antibodies showed high co-localization at PN0 suggesting that many OXT neurons in the PVN could co-express OXT-neurophysin-I and other intermediate OXT forms. Furthermore, the mature form of OXT co-exist with immature forms during the entire postnatal life (Otero-García et al., 2014; Grinevich et al., 2015).

Fig. 2. Immunohistofluorescence detection of OXT. Immunodetection of OXT-neurophysin I (associated with OXT prohormone) using the PS38 antibody (in red) and of OXT-intermediate forms using the VA10 antibody (in green). At E14.5, a few cells expressing the OXT-neurophysin I were detected (A), probably this signal corresponds to the OXT prohormone, but not to the OXT intermediate forms such as VA10 that appear at E16.5 (B). At birth time, many OXT neurons in the PVN co-express OXT-neurophysin I and the VA10 immature form (C). White arrows indicate the location of the third ventricle. *Figure adapted from Grinevich et al., 2015.*

These results indicate that the PS38 antibody used in this study recognizes an early variant of OXT. We further explored this possibility employing an antibody for recognizing full-length OXT (rabbit anti-OXT, Millipore AB911) (Fig. 3 in this revision, new **Supplementary Figure 3**). This antibody was able to recognize OXT at postnatal stages but no signal was observed in neonates, supporting the notion that PS38 antibody recognizes specific OXT variants expressed during early development.

Fig. 3. OXT immunohistofluorescence using rabbit anti-OXT Millipore Cat#AB911. PVN coronal sections and high-magnification at a range of developmental stages: PN0, PN2, PN14, PN21 and adult brain. *New Supplementary Figure 3*. Scale bar: **a-f**: 150 μ m; **a'-f'**: 50 μ m.

Moreover, we investigated OXT immunodetection with both antibodies in the adult brain replicating the results obtained in Otero-García et al., 2014. The two antibodies, mouse PS38 anti-OXT (OXT-neurophysin I) and rabbit anti-OXT (full-length oxytocin) co-expressed in all the analyzed regions in the adult brain (**Fig. 4** in this revision, new **Supplementary Figure 4**). These results are now indicated in lines 276-282.

Fig. 4. Double immunohistochemistry for detecting OXT. Images show the labeling of mouse PS38 anti-OXT (in green) and rabbit anti-OXT (Millipore Cat#AB911, in red). Coronal sections of adult mouse showing distinct brain regions: BNST (bed nucleus of the stria terminalis), PVN (paraventricular nucleus), SON (suprachiasmatic nucleus) and RCH (retrochiasmatic area). *New Supplementary Figure 4*. Scale bar: 100 μ m.

Since the rabbit anti-OXT (Millipore) is not able to detect OXT neurons at embryonic stages but both antibodies identify oxytocinergic neurons in the adult brain, we conclude that mouse anti-OXT (PS38) recognizes early OXT variants which may persist at postnatal stages (Otero-García et al., 2014; Grinevich et al., 2015). Therefore, the PS38 antibody can be used to identify the appearance of OXT-expressing cells during embryonic stages. However, we acknowledge that the actual degree of molecular maturation of the OXT peptide/s recognized by PS38 remains to be formally tested by directly assaying OXT amidation. Since these measurements are technically challenging and out of the scope of the present study, we have eliminated allusions to OXT maturation in the text.

Similar to OXT, AVP exists as an immature pre-pro-vasopressin-neurophysin II peptide that is further processed to generate the nonapeptide AVP and the carrier protein neurophysin II. A recent description of the AVP distribution in the adult mouse brain (Otero-García et al., 2014) employed the same antibody used in our study (Millipore PC234L) to identify the processed form. Interestingly, we are able to detect AVP expression with this antibody as early as E16.5, supporting the notion that AVP maturation occurs earlier than OXT maturation (Altstein & Gainer, 1988).

3. *Further, their statement beginning at line 160 is misleading as the authors are, in fact, not looking at the **small protein-like (what does this mean?) molecule Oxt, but its co-expressed protein neurophysin which is much larger.***

We apologize for this misunderstanding. We have now changed this by the word “*peptide*” (line 154) to avoid confusion.

4. *I am not convinced by the data that the Oxt-Cre line generated td-Tomato expression is necessarily indicative of natural Oxt gene expression. I would need to see **co-expression of Cre (or td-Tomato) and Oxt transcripts**. A possibility is that the Oxt-Cre line ectopically expresses Cre, at least temporally, out of synch from the Oxt locus.*

This is a valid concern that can be explained by the genetic strategy used to generate the transgenic mouse line used in our study. This line was generated by Wu et al., 2012 following a knock-in strategy to ensure the match of Cre and Oxt expression (see schematics of the recombination design in **Fig. 5**, adapted from the original work by Wu et al., 2012). Briefly, a DNA cassette containing an IRES sequence, a Cre coding sequence and a polyadenylation sequence were placed after the stop codon of the endogenous *Oxt* gene (**Fig. 5**). In this configuration, *Cre* and *Oxt* are transcribed as a single bicistronic mRNA and the IRES sequence allows simultaneous translation of both Oxt and Cre proteins from the same transcript and preventing the independent identification of OXT and Cre transcripts by *in situ* hybridization. Importantly, Oxt-Cre mice are viable and exhibit normal maternal behavior suggesting preserved expression patterns from the *Oxt* gene locus.

Similar to our study, to confirm appropriate co-expression of oxytocin and Cre recombinase at the protein level, Wu et al., crossed mice harboring *Oxytoxin-Ires-Cre* with a Td-tomato reporter strain which expresses Td-Tomato in a Cre-dependent manner (Ai9 mice). Immunostaining for oxytocin shows a near complete colocalization of dsRed (red) and oxytocin (green) restricted to the hypothalamic paraventricular (PVN) and supraoptic (SON) nuclei, where oxytocin is known to be expressed (**Fig 5**). Notably, Wu et al., found that approximately, 92% of cells exhibiting Oxt labeling also expressed Cre activity in the PVN (Wu et al., 2012), which is in a similar range to our data using the PS38 antibody (~ 96 % colocalization). Although OXT and tdTomato coexpression is almost complete in both studies, it is clear that some tdTomato-expressing neurons do not colocalize with Oxt (**Fig. 5B**, and **Figures. 8** and **9** in our manuscript).

Fig. 5. Oxytocin-IRES-Cre knock-in mice express Cre recombinase in OXT neurons. A) Cre recombinase was targeted just after the stop codon of the OXT gene using an internal ribosomal entry site (IRES). B) Immunohistochemistry for OXT inn brain slices of Oxytocin-IRES-Cre:Tdtomato reporter (Ai9) mice indicates that nearly all OXT-containing neurons (green) express Cre recombinase. *Figure adapted from Wu et al., 2012.*

This discrepancy may be explained by the fact that the level of oxytocin expression varies depending on the physiological state of the animal. Common events such as feeding has been shown to correlate with significant changes in OXT mRNA expression in the hypothalamus (Olszewski et al., 2010), so it is very likely that not all cells expressing tdTomato (expressed chronically once the stop codon is removed upon Cre recombination) are producing Oxt mRNA at the time of sacrifice. We have updated the manuscript including these considerations in the Results (lines 265-266; lines 276-282), Discussion (lines 348-358) and the Material and Methods section (lines 407-409).

5. In **figure 3**, please change the figure label to “Oxt+Avp”, not “Oxt-Avp.” At first I thought there was a distribution difference being displayed.

We apologize for this mistake that has been corrected in the new version of **Figure 3**.

Reviewer #2

1. Overall, this is an interesting paper that has utilized a state-of-the-art approach to evaluate the developmental timeline of oxytocin and vasopressin expression during development. Specifically, by using iDISCO+ coupled with light sheet microscopy the authors have provided a look into the development of these systems that has not been achieved before. There is no doubt that this work makes an important contribution to the field. However, as written, the paper oversimplifies these systems. Also, generally speaking, the manuscript seems **under cited**. For example, it is unlikely that oxytocin and vasopressin “**underlie neuropsychiatric disorders**”, this is an **overstatement**. Most often, these systems are dysregulated and/or affected by drug treatment in some individuals with specific neuropsychiatric disorders.

We appreciate the reviewer to point this out. Indeed, this sentence was poorly constructed and was confusing. This has now been corrected throughout the manuscript (*changes in the manuscript are indicated in red, lines 36-37, lines 81-82, lines 127-129*) to convey the notion that it is the dysregulation of these systems which may underlie some neuropsychiatric disorders. We also have increased the number of citations accordingly to the updated discussion.

2. There is almost no mention of **sex differences** in these systems, which is critically important given that there are identified sex differences in their development, as well as in adult animals, that are thought to be associated with their function.

We appreciate the reviewer’s comment on this important aspect of the oxytocinergic and vasopressinergic systems. As indicated by the reviewer, there is a vast body of work addressing the sexual dimorphism of the rodent hypothalamus (Liao et al., 2020; Smith et al., 2019; Scott et al., 2015; Otero-García et al., 2014; Rood et al., 2013; Yang et al., 2013; Bales et al., 2007; Uhl-Bronner et al., 2005; Simerly et al., 1998; Szot et al., 1993, among many others), however most of previous studies focused on OXT and AVP receptor expression, and used rat as the animal model.

More recently, a few articles explored these systems in the mouse brain (Otero-García et al., 2014; Otero-García et al., 2016; Sharma et al., 2019) exposing some sex differences such as the expression of the OXT receptor in the anteroventral periventricular nucleus (AVPV) exclusively in females (Sharma et al., 2019). However, sex differences regarding the expression of OXT and AVP peptides seem to be less obvious in the mouse brain, and a study by Otero-García et al., 2014 analysing the expression of OXT and AVP in the adult mouse brain did not reveal any significant differences between sexes. Given these results and the fact that our study employed the same antibodies used in Otero-García et al., 2014, we were discouraged to explore sex differences during development. Since we did not focus on sex differences we seldom referred to these findings in our previous version, however we have now updated the Discussion with a few sentences on this important aspect (lines 317-320) and expanded the References section accordingly (*new references 48-57*).

3. While the authors indicate that adult female brains were used, the embryonic and early postnatal timepoints make **no mention of the sex of the embryos/newborn pups used**. This is relevant because, for instance, Tamborski et al (2016) found no oxytocin mRNA to be present at E16.5 in the male mouse brain.

We apologize for this omission in the Methods. Since we encountered some difficulties to unambiguously identify sex in very young embryos and pups, the results at early developmental stages (E16.5, PN0 and PN7) were obtained from animals of both sexes. These technical details have been updated in the Material and Methods (lines 403-404).

- 4.-Lastly, how do the authors know that the oxytocin and vasopressin are expressed in neurons at E16.5 if no neuronal marker was utilized?

Early characterizations of the rodent hypothalamus revealed that OXT and AVP expression is restricted to different neuronal types (Vandesande et al., 1975; Mohr et al., 1988; Kiyama et al., 1990; Glasgow et al., 1999; Xi et al., 1999; Wang & Lufkin, 2000).

Fig. 6. Neuronal expression of OXT during early developmental stages. Representative images of immunohistofluorescence experiments to detect doublecortin (DCX, in red Abcam antibody) and OXT (in green, PS38 antibody) at E18.5. Despite DCX expression was high in cortical areas, no clear signal was detected in the PVN or SON in the same brain preparation.

Since these studies have not been revisited for several years, we performed a double immunostaining protocol to identify both OXT or AVP, and doublecortin (DCX) a marker of neuronal progenitors. We tested three antibodies against DCX (mouse anti-DCX Santa Cruz SC-271390; goat DCX (C-18) Santa Cruz SC-8066; rabbit anti-DCX Abcam ab18723) to identify neuronal progenitors in the developing hypothalamus. Surprisingly, all the antibodies tested provided negative results in the hypothalamus but exhibited a high signal in the cortical area (**Fig 6** shows a representative example of OXT and DCX immunostaining). These results suggest that hypothalamic neuronal progenitors are regulated by a distinct molecular machinery than progenitors in other brain regions. Previous work suggested that in fact the *Otp* homeobox gene plays an essential role in the specification of neuronal cell lineages in the developing mouse hypothalamus (Wang & Lufkin, 2000). Although these are interesting findings, we

did not further explore this issue since we consider it out of the scope of the current study. Nonetheless, we attempted to identify neurons using a different neuronal marker: MAP2 (mouse anti-MAP2 Chemicon MAB364) during early developmental stages (**Fig. 7A**). For these experiments we opted for culturing hypothalamic neurons from E16.5 mouse pups to obtain clear images of young neurons to avoid confounding results from the strong signal of MAP2, a very abundant protein which in our hands yields a very intense labelling in tissue slices. Immunolabeling of MAP2 and OXT or AVP at 4 DIV indicated that all OXT and AVP cells co-localize with the neuronal marker, as previously described. Representative images are shown in **Fig. 7A**. We also analysed the expression of OXT and AVP in astrocytes using a specific antibody against GFAP (mouse anti-GFAP Millipore MAB360), a common astroglial marker. We found no co-localization between GFAP and AVP in young cultures (top panel in **Fig. 7B**). Furthermore, we explored the expression of OXT in the adult hypothalamus, finding no OXT labelling in GFAP positive cells (bottom panel in **Fig. 7B**) indicating that the expression of these neuropeptides is restricted to neurons from early development to adulthood.

Fig. 7 Neuronal expression of OXT and AVP. **A)** Representative examples of neuronal marker MAP2 (in red) and OXT (in green, top panel) or AVP (in green, bottom panel) in young (4 DIV) cultured hypothalamic neurons obtained from E16.5 mouse pups. Co-localization between MAP2 and OXT or AVP was complete indicating that their expression is restricted to neurons. Scale bar represents 100 μ m. **B)** Representative images of the labelling obtained with an astrocyte marker (GFAP in red) and AVP (in green, top panel) in young hypothalamic cultures. Bottom panel shows the expression of OXT (in green) and GFAP (in red) in the adult PVN indicating that this neuropeptide is not expressed in this glial type. Scale bar represents 100 μ m.

5. Figure 1A is fuzzy and the nomenclature for AVP is not maintained in the supplemental.

We have updated **Figure 1** to improve the quality of the image and corrected the AVP nomenclature in the Supplementary Information.

References

Altstein M & Gainer H. Differential biosynthesis and posttranslational processing of vasopressin and oxytocin in rat brain during embryonic and postnatal development. *J. Neurosci.* 8: 3967–3977 (1988).

Althammer et al., Diversity of oxytocin neurones: Beyond magno- and parvocellular cell types? *J Neuroendocrinol.* doi: 10.1111/jne.12549. (2017)

Bales KL et al., Neonatal oxytocin manipulations have long-lasting, sexually dimorphic effects on vasopressin receptors. *Neuroscience.* 144:38-45 (2007)

Belenky M et al., Ultrastructural immunolocalization of rat oxytocin-neurophysin in transgenic mice expressing the rat oxytocin gene. *Brain Res* 583:279–286 (1992)

Ben-Barak Y et al., Neurophysin in the hypothalamo-neurohypophysial system. I. Production and characterization of monoclonal antibodies. *J Neurosci* 5:81–97 (1985)

Butovsky E et al., Chronic exposure to D9-tetrahydrocannabinol downregulates oxytocin and oxytocin-associated neurophysin in specific brain areas. *Mol Cell Neurosci* 31:795–804 (2006)

Glasgow E, et al., Single cell reverse transcription-polymerase chain reaction analysis of rat supraoptic magnocellular neurons: neuropeptide phenotypes and high voltage-gated calcium channel subtypes. *Endocrinology.* 140:5391–5401 (1999)

Grinevich V et al., Ontogenesis of oxytocin pathways in the mammalian brain: late maturation and psychosocial disorders. *Front Neuroanat.* 8, 164. (2015)

Hasan et al., A fear memory engram and its plasticity in the hypothalamic oxytocin system. *Neuron.* **103**, 133-146.e8 (2019)

Kiyama H, Emson PC. Evidence for the co-expression of oxytocin and vasopressin messenger ribonucleic acids in magnocellular neurosecretory cells: simultaneous demonstration of two neurohypophysin messenger ribonucleic acids by hybridization histochemistry. *Journal of Neuroendocrinology.* 2:257–259 (1990)

Liao PY et al., Mapping central projection of oxytocin neurons in unmated mice using Cre and alkaline phosphatase reporter. *Front. Neuroanat* <https://doi.org/10.3389/fnana.2020.559402> (2020)

Mason WT et al., Central release of oxytocin, vasopressin and neurophysin by magnocellular neurone depolarization: evidence in slices of guinea pig and rat hypothalamus. *Neuroendocrinology* 42, 311-22 (1986)

Mohr E et al., Expression of the vasopressin and oxytocin genes in rats occurs in mutually exclusive sets of hypothalamic neurons. *FEBS Lett.* 242:144–148 (1988)

Otero-García M et al., Extending the socio-sexual brain: arginine-vasopressin immunoreactive circuits in the telencephalon of mice. *Brain Struct Funct.* 219:1055-81 (2014)

Otero-García M et al., Distribution of oxytocin and co-localization with arginine vasopressin in the brain of mice *Brain Struct Funct* 221:3445-73 (2016)

- Olszewski PK et al., Molecular, immunohistochemical, and pharmacological evidence of oxytocin's role as inhibitor of carbohydrate but not fat intake. *Endocrinology*. 151:4736-44 (2010)
- Rao VV et al., The human gene for oxytocin-neurophysin I (OXT) is physically mapped to chromosome 20p13 by *in situ* hybridization. *Cytogenet Cell Genet* 61: 271–3 (1992)
- Rhodes CH et al., Immunohistochemical analysis of magnocellular elements in rat hypothalamus: distribution and numbers of cells containing neurophysin, oxytocin, and vasopressin. *J Comp Neurol* 198:45-64 (1981)
- Rood BD et al., Site of origin of and sex differences in the vasopressin innervation of the mouse (*Mus Musculus*) brain. *J Comp Neurol* 521:2321–2358 (2013)
- Scott N et al., A sexually dimorphic hypothalamic circuit controls maternal care and oxytocin secretion. *Nature*. 525:519–22 (2015)
- Sharma K et al., Sexually dimorphic oxytocin receptor-expressing neurons in the preoptic area of the mouse brain. *PLoS One*. 14(7):e0219784 (2019)
- Simerly RB. Organization and regulation of sexually dimorphic neuroendocrine pathways. *Behav Brain Res*. 92:195–203 (1998)
- Smith CJW et al., Comparing vasopressin and oxytocin fiber and receptor density patterns in the social behavior neural network: Implications for cross-system signaling, *Frontiers in Neuroendocrinology*, doi: <https://doi.org/10.1016/j.yfrne.2019.02.001> (2019)
- Sofroniew MW et al., The distribution of vasopressin-, oxytocin-, and neurophysin-producing neurons in the guinea pig brain. I. The classical hypothalamo-neurohypophyseal system. *Cell Tissue Res* 196:367-84 (1979)
- Stern JE et al., Secretion of vasopressin, oxytocin, and two neurophysins from rat hypothalamo-neurohypophyseal explants in organ culture. *Neuroendocrinology* 43, 252-8 (1986)
- Szot P., Dorsa D.M. Differential timing and sexual dimorphism in the expression of the vasopressin gene in the developing rat brain. *Brain Res Dev Brain Res* 73:177-83 (1993)
- Takahiro et al., A monoclonal antibody raised against a synthetic oxytocin peptide stains mouse hypothalamic neurons. *J Neuroendocrinology* doi.org/10.1111/jne.12815 (2020)
- Uhl-Bronner S et al., Sexually dimorphic expression of oxytocin binding sites in forebrain and spinal cord of the rat. *Neuroscience*. 135:147–54 (2005)
- Vandesande F, Dierickx K. Identification of the vasopressin producing and of the oxytocin producing neurons in the hypothalamic magnocellular neurosecretory system of the rat. *Cell Tissue Res*. 164:153–162. (1975)
- Wang & Lufkin. The murine Otp homeobox gene plays an essential role in the specification of neuronal cell lineages in the developing hypothalamus. *Developmental Biology* 227: 432-449 (2000)
- Whitnall MH et al., Neurophysin in the hypothalamo-neurohypophysial system. II. Immunocytochemical studies of the ontogeny of oxytocinergic and vasopressinergic neurons. *J Neurosci* 5:98–109 (1985)
- Wu Z et al., An obligate role of oxytocin neurons in diet induced energy expenditure. *PLoS One* 7(9): e45167 (2012)

Xi D, Kusano K, Gainer H. Quantitative analysis of oxytocin and vasopressin messenger ribonucleic acids in single magnocellular neurons isolated from supraoptic nucleus of rat hypothalamus. *Endocrinology*. 140:4677–4682 (1999)

Yang, C. F. et al. Sexually dimorphic neurons in the ventromedial hypothalamus govern mating in both sexes and aggression in males. *Cell* 153, 896–909 (2013)

Reviewers' comments:

Reviewer #1 (Remarks to the Author):

I appreciate the authors revisions but there are still the two major issues that they have not addressed, and I take blame if I was not clear in the original review.

I should mention that the term "variant" is inappropriate in the context of this study. I believe that the authors should use the term "precursor" instead.

1. My point was that if one is attempting to compare maturation of AVP and OXT (i.e, the appearance of the mature, final peptides), one shouldn't use an antibody against the mature peptide (AVP in this case) and another against the immature precursor (OXT in this case). Their data and others' show that the mouse anti-OXT antibody shows the earlier appearance of the precursor while the rabbit one shows the later appearance of the processed OXT peptide. What is the situation with the AVP antibody used? Does it recognize the precursor neurophysin or is able to recognize the peptide, whether released from the preprohormone, not fully processed or mature. Ideally, one would want to use two antibodies for each system, one that recognize the unprocessed preprohormone and one only the processed, fully mature peptide. I don't know what the situation is here.

2. Of course, OXT and RFP will be in the same cells. That was not my point. I specifically was wondering if there could be ectopic expression from the new OXT-Cre locus, perhaps in AVP or other cells. If natural OXT expression were a subset of a greater range of OXT-Cre expression, then RFP+OXT immunostaining would not answer that question. That is why I specifically asked about the expression of transcripts that can establish if ectopic expression occurs. One, should compare the expressions using a probe that targets the ires-Cre sequence and another probe to OXT transcript. I don't believe Wu et al has answered this question either as they also use a reporter line.

Reviewer #2 (Remarks to the Author):

While I still think that the work described in this manuscript is of value to the field, there continue to be significant issues with this paper, many of which are associated with the glossing over, or watering down of important concepts. For instance, in the introduction and discussion, the nuances of these systems, and their relationship to behavior, are not adequately acknowledged. One example, the new last line of paragraph 1 (lines 80-82), references a single paper from 2011 to make a sweeping statement about the role of these neuropeptides in neuropsychiatric disorders. Conclusions such as these simply cannot easily be drawn from the literature and certainly not from a single published paper. Another example is the reference to the Otero-Garcia paper (line 319), where it is suggested that there is an absence, or less prominence, of sexual dimorphisms in early development of both systems. As this particular paper only evaluated the AVP system, not the OXT system, this statement is misleading. Also, there are other papers, beyond the Otero-Garcia paper, which too suggest that the AVP system is not as sexually dimorphic in early development as the OXT system. It is just that the OXT system does, in fact, appear to be sexually dimorphic in early development in mice. It is also the case that the statement, "Given the technical difficult of identifying the sex at a very young age..." (lines 402-404), is fairly misleading. As the authors used advanced techniques to perform their work, a straightforward PCR for the SRY gene to determine the sex of the experimental animals does not seem too onerous. Unfortunately, by choosing to not sex the animals, the interpretation of the work is difficult, and its impact significantly lessened. I would also suggest that OXT and AVP being expressed in glia, cannot be ruled out as GFAP is not the only glial marker, S100B has some distinct expression patterns.

We highly appreciate reviewers' comments and suggestions, and are pleased to submit a revised version of our manuscript entitled "*Specification of oxytocinergic and vasopressinergic circuits in the developing mouse brain*". Please find below the point-by-point response:

Reviewer #1

1. *I should mention that the term "variant" is inappropriate in the context of this study. I believe that the authors should use the term "precursor" instead.*

We very much appreciate the suggestion. We completely agree that precursor is a more precise term in this context, thus, the revised manuscript includes the suggested change in lines **120, 287, 351, 361, 368, 370, 371, 375, 390** (new changes are marked in blue, previous changes marked in red).

2. *My point was that if one is attempting to compare maturation of AVP and OXT (i.e, the appearance of the mature, final peptides), one shouldn't use an antibody against the mature peptide (AVP in this case) and another against the immature precursor (OXT in this case). Their data and others' show that the mouse anti-OXT antibody shows the earlier appearance of the precursor while the rabbit one shows the later appearance of the processed OXT peptide. What is the situation with the AVP antibody used? Does it recognize the precursor neurophysin or is able to recognize the peptide, whether released from the preprohormone, not fully processed or mature. Ideally, one would want to use two antibodies for each system, one that recognize the unprocessed preprohormone and one only the processed, fully mature peptide. I don't know what the situation is here.*

We apologize for not elaborating this point sufficiently in our previous answer, in which we focused on addressing the specificity of the OXT antibodies during development. Pioneer studies on AVP and OXT maturation established that mature AVP can be identified as early as E14.5 and that AVP has fewer precursors than OXT (Alstein and Gainer, 1988). The antibody used in our work recognizes the mature peptide and similar to previous reports, revealed AVP expression in E16.5 samples. Since the expression of mature AVP occurs at early developmental stages, distinguishing between the mature form and the precursors should be done at a time window earlier than the time frame chosen for our study. This time frame was chosen after performing preliminary experiments with E12.5 and E14.5 brains which showed that not all the hypothalamic nuclei were fully formed, a necessary feature for our study aimed to identify potential differences in the temporal dynamics of the appearance of OXT and AVP cells in the distinct hypothalamic nuclei, regardless of the maturation of the peptides. Nonetheless, this is a very interesting question that we have addressed for OXT, which exhibits a complex maturation process within the time frame of our analysis, and for which we performed specific experiments. We have included a sentence pointing this out in the Methods section (**lines 434-435**)

3. *Of course, OXT and RFP will be in the same cells. That was not my point. I specifically was wondering if there could be ectopic expression from the new OXT-Cre locus, perhaps in AVP or other cells. If natural OXT expression were a subset of a greater range of OXT-Cre expression, then RFP+OXT immunostaining would not answer that question. That is why I specifically asked about the expression of transcripts that can establish if ectopic expression occurs. One, should compare the expressions using a probe that targets the *ires-Cre* sequence and another probe to OXT transcript. I don't believe Wu et al has answered this question either as they also use a reporter line.*

We apologize if including immunohistochemistry data in our response was confusing. The immunohistochemistry images were just to show the high percentage of colocalization between OXT- and RFP- expressing cells at the protein level in adult brains, but by no means an attempt to address this issue by performing immuno experiments.

Although these are very reasonable concerns, we are afraid that the requested experiments are likely to yield inconclusive results in the animal model used in this study. This transgenic line was generated following a knock-in strategy by inserting the Ires-Cre sequence after the stop codon of the OXT gene (Wu et al., *PLoS One* 2012) that results in a single transcript for Cre and OXT. Thus, designing probes to target two regions of the same transcript (one for OXT and another one for Ires) will yield a 1:1 expression ratio. Even in the case that the two probes show different results, a likely explanation could be a difference in the efficacy with which the two probes recognize their correspondent regions of the (same) transcript. This is a technological limitation that we fear may start a long troubleshooting that will end when we obtain the expected 1:1 ratio. Conclusions regarding ectopic expression induced by the introduction of the Cre locus should be addressed testing natural OXT in the same animal. Then, it would be plausible to design two probes to detect the Ires-Cre region and the natural OXT transcript to compare the impact of the new Cre locus in OXT expression, but unfortunately this is not the situation in these homozygous knock-in animals in which natural OXT has been replaced by the Ires-Cre/OXT transcript. This explanation is included in the Methods section (**lines 412-414**). Furthermore, experiments performed in two mouse lines (WT with natural OXT and transgenic OXT-Cre) will be inconclusive since basic internal controls cannot be carried out. Thus, we are afraid that embarking on these experiments is unlikely to provide a meaningful outcome due to the genetic design of the mouse line.

Reviewer #2

“... in the introduction and discussion, the nuances of these systems, and their relationship to behavior, are not adequately acknowledged. One example, the new last line of paragraph 1 (lines 80-82), references a single paper from 2011 to make a sweeping statement about the role of these neuropeptides in neuropsychiatric disorders. Conclusions such as these simply cannot easily be drawn from the literature and certainly not from a single published paper...”

We apologize for attempting to summarize the extensive literature addressing the role of OXT and AVP systems in regulating behavior and the potential consequences of their dysregulation in just a few papers. This solely responded to concerns regarding space limitations. We have now included more articles and review papers focused on the role of OXT and AVP in modulating natural social behaviors (Caldwell et al., 2012; Caldwell, 2017; Hammock and Young, 2006; Lim and Young, 2006; McCarthy et al., 2009; de Vries et al., 2012), papers that link alterations of these systems to disorders such as autism (Lee et al., 2008; Crawley et al., 2007; Winslow & Insel, 2002; Green et al., 2001; Meyer-Lindenberg et al., 2009), social anxiety disorder (Meyer-Lindenberg et al., 2011), and schizophrenia (Caldwell et al., 2009; Lee et al., 2005; Goldman et al., 2008). Together, these papers cover many of the work that contributed to the field over the years. New sentences and references indicating the broad modulatory function of these systems can be found in the Abstract (**line 35**) and Introduction (**lines 75-84**) (new changes are marked in blue, previous changes in red).

Another example is the reference to the Otero-Garcia paper (line 319), where it is suggested that there is an absence, or less prominence, of sexual dimorphisms in early development of

both systems. As this particular paper only evaluated the AVP system, not the OXT system, this statement is misleading. Also, there are other papers, beyond the Otero-Garcia paper, which too suggest that the AVP system is not as sexually dimorphic in early development as the OXT system. It is just that the OXT system does, in fact, appear to be sexually dimorphic in early development in mice.

We have modified this statement to make clear that there is evidence indicating that OXT and AVP exhibit differences in sexual dimorphism in the developing mouse brain, and cite the work of Tamborski et al., 2016 that clearly shows this scenario (lines 319-321).

It is also the case that the statement, “Given the technical difficult of identifying the sex at a very young age...” (lines 402-404), is fairly misleading. As the authors used advanced techniques to perform their work, a straightforward PCR for the SRY gene to determine the sex of the experimental animals does not seem too onerous. Unfortunately, by choosing to not sex the animals, the interpretation of the work is difficult, and its impact significantly lessened.

Although our study was not originally designed to address sex-differences, we acknowledge the relevance of this aspect. So, in order to provide more detailed information about our methods, we attempted to perform PCR experiments using DNA extracted from the brains used in our study, which was the only material left from these animals. Unfortunately, the DNA obtained from the clarified tissue did not have enough quality to detect the SRY or IL3 genes using the oligos described in Tamborski et al., 2016. Results from the agarose gels can be found in **Figure 1A**. Females are expected to show a single band for IL3 (543 bp), whereas the presence of an IL3 band and a SRY band (401 bp) is characteristic of a male. In the first lane, we loaded the PCR product from fresh DNA obtained from a male mouse (control) showing the two expected bands (543 bp and 401 bp). The second and third lanes show the PCR products from adult brains conserved in DBE after the iDISCO protocol (male and female adult mouse used in the study). Since the samples conserved in DBE did not show clear bands, we tried the PCR using a range of annealing temperatures: 55 °C, 60 °C and 65 °C (**Figure 1B**). For these experiments, we used DNA from a control male (C) and DNA from an adult male processed with the iDISCO protocol (M). As shown in **Figure 1B**, the new temperatures did not improve the results.

Fig. 1. Sex determination by PCR.

Given these results, we have updated the Methods section adding a new sentence indicating that one limitation of the study is the use of a mixture of male and female brains to obtain the results corresponding to early developmental stages (E16.5, PN0 and PN7) (lines **407-409**).

I would also suggest that OXT and AVP being expressed in glia, cannot be ruled out as GFAP is not the only glial marker, S100B has some distinct expression patterns.

As suggested, we performed additional colocalization experiments using an alternate glial marker S100B to complement our GFAP data. We used two commercial S100B antibodies: mouse anti-S100B (Sigma S2532) and mouse anti-S100B (ThermoScientific MS-296-P1) with similar results (**Figure 2**). Although, we were not able to identify OXT-expressing glial cells in our experimental conditions (**Figure 2**), we agree that glial cells are highly heterogeneous, thus it is plausible that OXT and AVP may be expressed on specific glial types, perhaps during particular developmental periods, or in response to specific stimuli. Prompted by reviewer's suggestion we searched for articles describing a potential role of OXT in regulating glial function, a field that we did not know that well, and for which we thank the reviewer for pointing it out. This is an interesting research line addressing oxytocin-dependent regulation of glial function in the context of brain homeostasis and social behavior, thus, we now include a brief mention of this alternative function of OXT in the Discussion (lines **392-394**), and cite two articles as examples of astroglial and microglial regulation by OXT (Wang et al., 2009; Loth & Donaldson, 2021).

Fig 2. S100B expression in hypothalamic neurons. Brain coronal sections show the PVN (**a-c, d-f**) and SON (**d-i**) of an adult female. Immunohistofluorescence of OXT (in red) and S100B (in green). Two S100B antibodies were used: mouse anti-S100B (Sigma S2532; **a-c**) and mouse anti-S100B (ThermoScientific MS-296-P1; **d-i**). Abbreviations: PVN, paraventricular nucleus; SON, supraoptic nucleus. Scale bar: 50 μ m.

References

Altstein M & Gainer H. Differential biosynthesis and posttranslational processing of vasopressin and oxytocin in rat brain during embryonic and postnatal development. *J. Neurosci.* 8: 3967–3977 (1988).

Caldwell, H., K. Oxytocin and Vasopressin: Powerful Regulators of Social Behavior. *Neuroscientist.* 23, 517-28 (2017).

Caldwell, H., K. Neurobiology of sociability. *Adv Exp Med Biol.* 739,187-205 (2012).

- Caldwell, H., K., Stephens, S., L. & Young, W., S., 3rd. Oxytocin as a natural antipsychotic: a study using oxytocin knockout mice. *Mol Psychiatry*. **14**, 190–96 (2009)
- Consiglio, A., R. & Lucion, A., B. Lesion of hypothalamic paraventricular nucleus and maternal aggressive behavior in female rats. *Physiol Behav*. **59**, 591–96 (1996)
- Crawley, J., N., Chen, T., Puri, A., et al. Social approach behaviors in oxytocin knockout mice: comparison of two independent lines tested in different laboratory environments. *Neuropeptides*. **41**, 145–63 (2007)
- de Vries, G., J., Veenema, A., H. & Brown, C., H. Vasopressin and oxytocin: keys to understanding the neural control of physiology and behaviour. *J Neuroendocrinol*. **24**, 527 (2012).
- Goldman, M., Marlow-O'Connor, M., Torres, I., et al. Diminished plasma oxytocin in schizophrenic patients with neuroendocrine dysfunction and emotional deficits. *Schizophr Res*. **98**, 247–55 (2008)
- Green, L., Fein, D., Modahl, C., et al. Oxytocin and autistic disorder: alterations in peptide forms. *Biol Psychiatry*. **50**, 609–13 (2001)
- Hammock, E., A. & Young, L., J. Oxytocin, vasopressin and pair bonding: implications for autism. *Philos Trans R Soc Lond B Biol Sci*. **361**, 2187-98 (2006).
- Lee, H., J., Caldwell, H., K., Macbeth, A., H., et al. A conditional knockout mouse line of the oxytocin receptor. *Endocrinology*. **149**, 3256–63 (2008)
- Lee, P., R., Brady, D., L., Shapiro, R., A., et al. Social interaction deficits caused by chronic phencyclidine administration are reversed by oxytocin. *Neuropsychopharmacology*. **30**, 1883–94 (2005)
- Lim, M., M. & Young, L., J. Neuropeptidergic regulation of affiliative behavior and social bonding in animals. *Horm Behav*. **50**, 506-17 (2006).
- Loth, M., K. & Donaldson, Z., R. Oxytocin, Dopamine, and Opioid Interactions Underlying Pair Bonding: Highlighting a Potential Role for Microglia. *Endocrinology*. **1**, 162, bqaa223. doi: 10.1210/endo/bqaa223 (2021).
- McCarthy, M., M, Wright., C., L. & Schwarz, J., M. New tricks by an old dogma: mechanisms of the organizational/activational hypothesis of steroid-mediated sexual differentiation of brain and behavior. *Horm Behav*. **55**, 655-65 (2009).
- Meyer-Lindenberg, A., Kolachana, B., Gold, B., et al. Genetic variants in AVPR1A linked to autism predict amygdala activation and personality traits in healthy humans. *Mol Psychiatry*. **14**, 968-75 (2009)
- Tamborski, S., Mintz, E., M. & Caldwell, H., K. Sex Differences in the Embryonic Development of the Central Oxytocin System in Mice. *J Neuroendocrinol*. **28**, doi: 10.1111/jne.12364.
- Wang, Y., F. & Hatton, G., I. Astrocytic plasticity and patterned oxytocin neuronal activity: dynamic interactions. *J Neurosci*. **29**, 1743-54 (2009)

Winslow, J.T. & Insel, T., R. The social deficits of the oxytocin knockout mouse. *Neuropeptides*. **26**, 221-229 (2002)

REVIEWERS' COMMENTS:

Reviewer #1 (Remarks to the Author):

The authors have addressed points 1 and 2, but not 3. The solution is quite simple - just compare Oxt cell locations in the WT compared to the homozygous KI mice. If there is ectopic expression or aberrant expression, it will be apparent. They could also compare Hets to Homozygous to see if there is a drop of expression.

Reviewer #2 (Remarks to the Author):

I greatly appreciate the earnest attempts made to address some of my comments. While I do think it is unfortunate to not have the sex-specific information, the manuscript overall is much improved and the data remain interesting.

We highly appreciate the comments of the reviewers and editorial suggestions. We are pleased to submit a revised version of our manuscript entitled "*Specification of oxytocinergic and vasopressinergic circuits in the developing mouse brain*". Please find below the point-by-point response:

1) Please ensure that you upload the attached checklist when submitting your revision:

Check list is uploaded as a "Related Manuscript File"

2) Please ensure all author information is provided as in previous versions of the manuscript. Please follow the guidance in the attached checklist and avoid the use of initials:

We have revised the names of the authors as indicated. We would like to have first author Pilar Madrigal also as a corresponding author, however we have not been able to change this in the online submission system. Thus, we request your assistance to make this change in the final version of the manuscript.

3) Please ensure that the legends for your supplementary figures are in your supplementary file rather than in the main manuscript file. Please see attached checklist for guidance regarding your video legends:

Legends for supplementary figures are included in the supplementary file. Video legends are included in the right column of the Editorial Requests Table.

4) Please check the formatting of figure 4 – specifically the formatting of the left-hand borders for panels d, l and u. Also, regarding figure 4, there are too many panels. We allow up to 10 main figures so you may want to split this into 2 figures (and ensure that the references to the new figures in the text are updated accordingly) or move some of the figure to supplementary:

We have corrected the format as indicated. Figure 4 has been divided in two panels (Figure 4 and new Supplementary Figure 3).

5) Please provide data points on the graphs in figure 5 and ensure you provide the source data (see checklist):

Data points have been added to the figure.

6) Please re-label your panels in figure 8. For example, instead of a' and a'' please use ai and aii. Please ensure the text is updated accordingly. Please also alter the labelling of the panels in figures S3 and S4 in the same way:

We have re-labeled the panels for these figures and change the text accordingly.

7) Please ensure that a scale bar is included in figure S4:

A scale bar has been included in the figure.

8) Please ensure that antibody dilutions are provided in your methods:

This information is now included in the methods section.